# MULTIMODAL MASKED POLYMER AUTOENCODER FOR UNIFIED POLYMER INFORMATICS

## ABSTRACT

Recent advances in large-scale sequence modeling have opened up new opportunities for polymer informatics, enabling both property prediction from structures and inverse design of structures from desired properties. Most existing approaches, however, model these tasks as separate mappings, limiting their flexibility and robustness. We propose a multimodal representation learning framework that unifies diverse polymer informatics tasks within a single model. Our approach treats each property or structural element as an individual submodality and introduces an information-theoretic objective that balances informativeness across arbitrary subsets of modalities. The resulting Multimodal Masked Polymer Autoencoder (MMPAE) serves as an end-to-end foundation model, supporting both cross-modal generation and retrieval. Extensive experiments on large polymer datasets show that MMPAE not only surpasses strong task-specific baselines under realistic missing-value conditions, but also provides a flexible platform for diverse downstream applications with a unified architecture.

## 1 INTRODUCTION

Polymers play a central role in modern materials science, foundational to applications ranging from polymer electrolytes for batteries and fuel cells (Wang et al., 2020; Xie et al., 2021) to organic optoelectronics (St John et al., 2019; Munshi et al., 2021) and energy storage devices (Luo et al., 2018; Hu et al., 2020). Despite their ubiquity, discovering and designing new polymers remains a formidable challenge. The chemical design space, which spans monomers, copolymer compositions and processing conditions, is effectively infinite, making exhaustive exploration infeasible. Traditional trial-and-error synthesis and physics-based simulations cannot efficiently navigate such complexity, motivating the development of data-driven approaches to polymer informatics for accelerating materials discovery.

Recent advances in large-scale sequence modeling, inspired by large language models (LLMs), have enabled powerful polymer representations by treating chemical data as token sequences. These models demonstrate strong performance in inverse design (Qiu & Sun, 2024; Dobberstein et al., 2024; Chang & Ye, 2024a), and property prediction (Xu et al., 2023; Kuenneth & Ramprasad, 2023). Building on this progress, multimodal frameworks increasingly integrate integrate complementary structural information: MMPolymer (Wang et al., 2024) combines PSMILES (Kim et al., 2018) with 3D conformations, Uni-Poly (Huang et al., 2025) merges PSMILES, 2D graphs, 3D geometries, and contextual descriptors. Extending beyond structural integration, recent research advocates treating numerical properties as distinct input modalities, utilizing them as foundational tabular data (Costa et al., 2025), or distinct input categories (Ni et al., 2025; Zhou et al., 2025) to provide essential physically grounded information complementary to topological structure.

Despite these advances, most existing approaches remain constrained by their underlying architectural assumptions. For example, inverse-design frameworks typically formulate polymer generation as a one-way mapping from property to structure. This unimodal modeling neglects the intrinsically bidirectional and multifaceted relationship between polymer structures and their properties, thereby resulting in sub-optimal performance (Wang et al., 2024). Multimodal methods aim to address this limitation, but they usually operate at the modality level, aligning entire property sets with structural representations while overlooking the finer granularity within each modality (e.g. PSMILES tokens or individual property values). Such coarse-grained formulations fails to capture the intricate interplay between local structural motifs (e.g. functional group, backbone chain) and the collec-

tive physicochemical behaviors of polymers. As a result, they often struggle to support real-world scenarios where the input conditions are sparse or incomplete.

In this work, we present a multimodal representation learning framework that unifies cross-modal generation and retrieval of polymer structures and their physicochemical properties within a single model. We introduce the Multimodal Masked Polymer Autoencoder (MMPAE), a Transformer-based architecture that encodes both polymer structures and properties by treating each attribute as an individual submodality. While a straightforward masked reconstruction objective can learn joint representations, we show that it fails to balance mutual information (MI) across different input subsets, leading to under-represented modalities. To overcome this limitation, we incorporate a hierarchical mixture-of-experts mechanism that reweights unimodal and joint contributions. Our theoretical analysis and empirical results demonstrate that this approach yields more balanced representations and, in particular, significantly improves cross-modal task performance. Our contributions are threefold:

1. **Unified multimodal foundation model.** We propose a principled framework that models each attribute within a modality as an individual submodality and implements a single Transformer-based model capable of cross-modal generation and retrieval while handling arbitrary missing values.

2. **Information-theoretic regularization.** We introduce a hierarchical mixture-of-experts encoder and an MI objective which jointly encourage balanced informativeness across input subsets and align unimodal and complete-submodality representations in the latent space.

3. **Comprehensive evaluation.** Extensive experiments on various tasks using large-scale polymer datasets show that MMPAE outperforms both existing multimodal approaches and strong task-specific baselines in property prediction, inverse design, and cross-modal retrieval, particularly under realistic missing-value conditions.

By explicitly modeling polymers as a collection of interrelated submodalities, our approach provides a flexible foundation for polymer informatics, bridging structure and property within a single end-to-end model.

## 2 RELATED WORK

**Polymer property prediction** Transformer-based sequence models have recently been applied to polymer property prediction. PolyBERT (Kuenneth & Ramprasad, 2023), one of the first large-scale models tailored for polymers, extends the DeBERTa architecture (He et al., 2020) and is pre-trained on PSMILES representations of polymers using a masked language modeling objective. This enables PolyBERT to learn polymer embeddings that can be fine-tuned for downstream property prediction tasks. TransPolymer (Xu et al., 2023) similarly adopts the RoBERTa framework (Liu et al., 2019) to obtain meaningful polymer embeddings. These approaches highlight the effectiveness of pre-training on polymer sequences for capturing structure–property relationships.

**Polymer inverse design** Recent advances in polymer inverse design have leveraged transformer and diffusion architectures to generate polymer structures conditioned on target properties. Poly-TAO (Qiu & Sun, 2024) introduces a transformer-based foundation model that incorporates structural constraints, such as aromatic ring, and molecular weight, achieving higher validity and stronger property–structure alignment than earlier reconstruction or translation approaches. Llamol (Dobberstein et al., 2024) extends the Llama-2 architecture (Touvron et al., 2023) to jointly condition on target properties, using stochastic dropping during training to enable flexible generation while preserving structural validity or property diversity. LDMol (Chang & Ye, 2024a) employs a transformer-based diffusion model to bridge continuous and discrete molecular spaces, combining diffusion with contrastive learning on SMILES pairs to generate molecules conditioned on natural language property descriptions, such as functional groups, and substructures. Collectively, these methods highlight the potential of transformer- and diffusion-based pre-trained models for property-to-structure generation, but they still cast inverse design as a one-way mapping from properties to polymer structures.

**Multimodal approaches** Recent studies have begun to employ multimodal architecture to model the polymer structure-property relationship. MMPolymer (Wang et al., 2024) employs a dual-stream design with a Transformer for 1D sequences and a GNN for 3D conformations using masked language modeling with contrastive alignment to integrate sequence and geometric infor-

mation. PolyNC (Qiu et al., 2024) enables unified prediction of various properties via natural-language descriptions with a text-to-text transformer. SPMM (Chang & Ye, 2024b) introduces a multi-modal foundation model that fuses molecular structure and property information with cross-attention and bidirectional objectives, enabling simultaneous property prediction and inverse design. Uni-Poly (Huang et al., 2025) further extends multimodality by integrating SMILES, 2D graphs, 3D geometries, and polymer-context captions generated by large language models, aligning these heterogeneous sources through contrastive pre-training for superior performance on diverse property prediction tasks. Complementary modalities beyond structural inputs have also been utilized. Costa et al. (2025) encode composition text, structural imaging, and numerical properties using modality-specific encoders and integrate them into a shared latent space for improved prediction. Zhou et al. (2025); Ni et al. (2025) develop multimodal–multitask frameworks that simultaneously predict degradability, mechanical, and other polymer properties, either by fusing molecular graphs with physicochemical descriptors Zhou et al. (2025) or by combining nuclear magnetic resonance spectral features with thermal property inputs Ni et al. (2025). These methods commonly demonstrate the benefits of combining complementary data sources.

## 3 METHOD

Let $x = [x_1, ..., x_{|X|}]$ be PSMILES representation of a polymer and $y = [y_1, ..., y_{|Y|}]$ be its properties such as degradation temperature, heat capacity, density, and gas permeability whose values are continuous. We thus represent a polymer using two modalities, $x$ and $y$, where $|X|$ and $|Y|$ correspond to the numbers of submodalities, i.e., PSMILES tokens and individual properties, respectively. The concatenation of the two modalities is written as $xy = [x_1, ..., x_{|X|}, y_1, ..., y_{|Y|}] \sim p_D(\cdot)$, where $p_D$ is the empirical data distribution. Let $s \in \mathcal{P}(xy)$ denote a subset of the submodalities of $xy$, where $\mathcal{P}$ is the power set. When needed, we use $s_x \in \mathcal{P}(x)$, $s_y \in \mathcal{P}(y)$ to indicate unimodal subsets, and $s_{xy} \in \mathcal{P}(xy) \setminus (\mathcal{P}(x) \cup \mathcal{P}(y))$ to denote multimodal subsets spanning both modalities. Given this multimodal representation of polymer data, our goal is to train a single model that supports cross-modal generation (e.g. PSMILES to property, and property to PSMILES) and retrieval (e.g. retrieving the top-K PSMILES that best match a given set of properties, or retrieving properties for a given PSMILES), while remaining robust to missing submodalities. To this end, we propose a multimodal representation learning framework that drives the model to capture informative and robust representations across all submodalities. We first introduce a Transformer-based autoencoder capable of processing multiple modalities and trained with a simple masked-reconstruction objective (Section 3.1). Next, we provide an information-theoretic analysis that reveals the limitations of using the reconstruction objective (Section 3.2). Building on this insight, we present an auxiliary regularization objective that further enhances the learned representations, ensuring robustness to missing submodalities while preserving full informativeness (Section 3.3).

### 3.1 TRAINING A TRANSFORMER AUTOENCODER WITH MASKING SUBMODALITIES

To capture interactions across modalities in the unified representation, we adopt an early-fusion strategy. Specifically, submodal inputs from the PSMILES and the property modalities are mapped into a common embedding space so that a single Transformer encoder ($\theta$) can jointly process them. Following PolyBERT (Kuenneth & Ramprasad, 2023), we tokenize each string in PSMILES sequences and embed each token $x_i$ with positional embeddings to preserve sequence order. For properties, each property value $y_j$ is projected through an independent linear layer following Llamol (Dobberstein et al., 2024). We further incorporate a modality-shared embedding as in M3AE (Geng et al., 2022) so that the encoder can distinguish between modalities. The PSMILES and property embeddings are then concatenated together with a [CLS] head token, and the entire sequence is fed to the Transformer encoder. Following the Vision Transformer (ViT) design, we use only the encoder output associated with the [CLS] token as the aggregated representation.

Given this serialized embedding, the Transformer-based joint encoder $p_\theta^J$ is trained to encode the multimodal input under random submodality masking. When each submodality is independently masked with probability 0.5, the encoder can be interpreted as a Mixture of Experts (MoE) in which each "expert" corresponds to a subset of submodalities $s$, although all experts share the same set of parameters $\theta$. Formally,

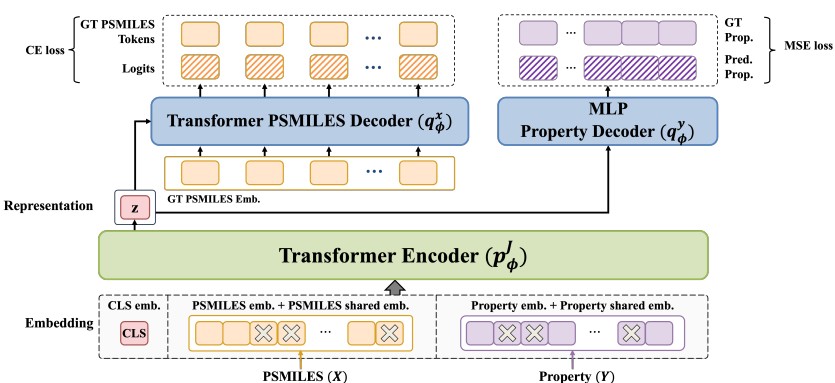

Figure 1: Training process of MMPAE with reoncstruction objective (Equation 2).

$$p_\theta^J(z \mid xy) = \frac{1}{2^{|X|+|Y|}} \sum_{s \in \mathcal{P}(xy)} p_\theta^s(z|s), \tag{1}$$

where $p_\theta^s$ denotes the Transformer subset encoder for $s$.

It is important to note that $p_\theta^J$ differs from the encoders used in typical Masked Autoencoders (He et al., 2022; Geng et al., 2022), which operate with a fixed number of input submodalities and therefore cannot learn from the full range of possible input combinations. In contrast, $p_\theta^J$ is inspired by the MoPoE encoder (Sutter et al., 2021), a late-fusion architecture that explicitly models representations for every combination of modalities with equal weights. This design allows the model to capture both shared and modality-specific information that would be lost if only a limited subset of combinations were considered. Our encoder extends this idea from modalities to *submodalities* by masking each submodality independently and aggregating the resulting subsets in an early-fusion manner. A detailed comparison between our encoder and multimodal VAEs, including MoPoE-VAE, is provided in Section A.2.

In addition to the joint encoder, we attach two decoders: an autoregressive Transformer decoder $q_\phi^x$ for PSMILES reconstruction and an MLP decoder $q_\phi^y$ for property prediction. The complete model, illustrated in Figure 1, is referred to as the Multimodal Masked Polymer Auto-Encoder (MMPAE).

### 3.2 An Information-Theoretic Analysis of MMPAE

A straightforward training objective is the masked reconstruction (Devlin et al., 2019; He et al., 2022; Geng et al., 2022), which maximizes the likelihood of recovering both modalities from the latent representation $z \sim p_\theta^J$ (Equation 2). Sampling representations from all subset combinations with equal probability trains the model to infer complete information from partial inputs. However, because the objective averages over exponentially many subsets, it provides little incentive for single-modality representations, which can limit downstream tasks such as cross-modal generation or retrieval.

We can understand this limitation from an information-theoretic perspective. Maximizing the reconstruction objective also maximizes Mutual Information (MI) between the joint representation $z$ and all submodalities, since it provides a lower bound on MI:

$$\underbrace{H(XY)}_{\text{Entropy (constant)}} + \underbrace{\mathbb{E}_{p_D(xy)p_\theta^J(z|xy)}\left[\log q_\phi(xy \mid z)\right]}_{\text{Reconstruction objective}} \leq \underbrace{I_\theta(Z; XY)}_{\text{MI objective}}, \tag{2}$$

where $q_\phi$ is a factorized joint decoder s.t. $q_\phi(xy \mid z) = q_\phi^x(x|z)q_\phi^y(y|z)$.

Importantly, because the joint encoder $p_\theta^J$ is a uniform mixture of subset experts, the MI term in Equation 2 can be upper-bounded by MI terms over all possible subsets, each measuring the dependence between a subset and its latent representation as formalized below:

**Proposition 1.** *Given the joint encoder $p_\theta^J$ defined as Equation 1,*

$$I_\theta(Z; XY) \leq \frac{1}{2^{|X|+|Y|}} \sum_{S \in \mathcal{P}(XY)} I_\theta(Z_S; S).$$

*Proof.* See Section A.1 in the supplementary material. □

Proposition 1 shows that optimizing the reconstruction objective in (Equation 2) implicitly maximizes the MI of all subsets of submodalities $I_\theta(Z_S; S)$ with equal weight $\frac{1}{2^{|X|+|Y|}}$. Such uniform weighting can create unbalanced informativeness among modality-specific subsets and cross-modal subsets, which becomes clear when we categorize the subset encoders into three types of experts:

$$p_\theta^s(z|s) = \begin{cases} p_\theta^{s_x}(z|s_x) & \text{if } s = s_x \quad \text{s.t. } s \in \mathcal{P}(x) \\ p_\theta^{s_y}(z|s_y) & \text{if } s = s_y \quad \text{s.t. } s \in \mathcal{P}(y) \\ p_\theta^{s_{xy}}(z|s_{xy}) & \text{if } s = s_{xy} \quad \text{s.t. } s \in \mathcal{P}(xy) \setminus (\mathcal{P}(x) \cup \mathcal{P}(y)) \end{cases} \tag{3}$$

This facilitates an alternative view of $p_\theta^J$ that exposes three low-level MoEs: two Mixtures of Unimodal Experts (MoUE) conditioned on the individual modalities ($*$ and $**$) and the Mixture of Multimodal Experts (MoME) conditioned on the joint modality ($***$):

$$p_\theta^J(z \mid xy) = \frac{1}{2^{|Y|}} \underbrace{p_\theta^x(z \mid x)}_{\text{MoUE of } x(*)} + \frac{1}{2^{|X|}} \underbrace{p_\theta^y(z \mid y)}_{\text{MoUE of } y(**)} + \left(1 - \frac{2^{|X|} + 2^{|Y|}}{2^{|X|+|Y|}}\right) \underbrace{p_\theta^{xy}(z \mid xy)}_{\text{MoME of } xy(***)}, \tag{4}$$

$$\text{where} \quad p_\theta^x(z \mid x) = \frac{1}{2^{|X|}} \sum_{s_x \in \mathcal{P}(x)} p_\theta^{s_x}(z|s_x) \ (*), \quad p_\theta^y(z \mid y) = \frac{1}{2^{|Y|}} \sum_{s_y \in \mathcal{P}(y)} p_\theta^{s_y}(z|s_y) \ (**),$$

$$p_\theta^{xy}(z \mid xy) = \frac{1}{2^{|X|+|Y|} - 2^{|X|} - 2^{|Y|}} \sum_{\substack{s_{xy} \in \mathcal{P}(xy) \\ \setminus (\mathcal{P}(x) \cup \mathcal{P}(y))}} p_\theta^{s_{xy}}(z|s_{xy}) \ (***).$$

The MI between $z$ and $xy$ is thus upper-bounded by a weighted sum of three MI terms involving $z^x \sim p_\theta^x$, $z^y \sim p_\theta^y$, and $z^{xy} \sim p_\theta^{xy}$ are the representations produced by the low-level MoEs:

$$I_\theta(Z; XY) \leq \frac{1}{2^{|Y|}} \underbrace{I_\theta(Z_X; X)}_{\star} + \frac{1}{2^{|X|}} \underbrace{I_\theta(Z_Y; Y)}_{\star\star} + \left(1 - \frac{2^{|X|} + 2^{|Y|}}{2^{|X|+|Y|}}\right) \underbrace{I_\theta(Z_{XY}; XY)}_{\star\star\star}. \tag{5}$$

Each MI term lower-bounds the sum of MI values for the corresponding intra-modality subsets, e.g., $I_\theta(Z_X; X) \leq \frac{1}{2^{|X|}} \sum_{S_X \in \mathcal{P}(X)} I_\theta(Z_{S_X}; S_X)$ ($\star$), and analogously for $I_\theta(Z_Y; Y)$ ($\star\star$) and $I_\theta(Z_{XY}; XY)$ ($\star\star\star$).

Equation 5 highlights a key limitation of optimizing with the reconstruction objective in Equation 2. For downstream tasks such as cross-modal generation and retrieval, informativeness of single-modality representations, $z_x$ and $z_y$, are critical. However, their coefficients $2^{-|X|}$ and $2^{-|Y|}$ decay exponentially with the number of submodalities, driving the MI of $z_x, z_y$ toward zero. Consequently, they can fail to extract meaningful information from single modalities.

To overcome this imbalance in informativeness, the next section introduces mechanisms that explicitly increase the weights of the unimodal MI terms $I_\theta(Z_X; X)$ and $I_\theta(Z_Y; Y)$, ensuring stronger single-modality representations.

### 3.3 MI Maximization via Hierarchical Mixture of Experts

To increase the coefficients of the single-modality MI terms, we introduce a Hierarchical Mixture of Experts (HMoE) as the joint encoder $p_\theta^{\text{HMoE}}$. This encoder allows explicit control over the weights (hyperparameters $\lambda^x, \lambda^y$ and $\lambda^{xy}$ which sum to one) of the low-level mixtures:

$$p_\theta^{\text{HMoE}}(z \mid xy) = \lambda^x \cdot p_\theta^x(z \mid x) + \lambda^y \cdot p_\theta^y(z \mid y) + \lambda^{xy} \cdot p_\theta^{xy}(z \mid xy). \tag{6}$$

The formulation is hierarchical because each low-level expert is first constructed as a mixture over masked submodalities within its designated subset, and these experts are subsequently aggregated by the top-level mixture. This two-stage construction yields a hierarchy from submodality mixtures

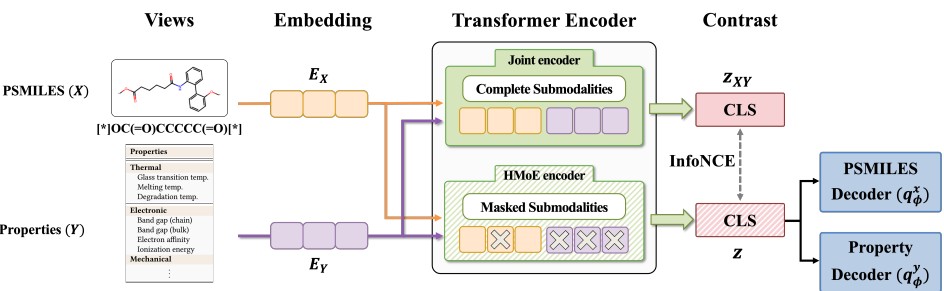

Figure 2: Training process of MMPAE with Equation 9. The joint encoder produces a representation from complete submodalities, while the HMoE encoder produces a masked counterpart. The masked representation is used for reconstruction, and both representations are used in the InfoNCE alignment. For simplicity, the detailed decoding pipeline is omitted.

to joint modality selection. Sampling from $p_\theta^{\text{HMoE}}$ proceeds in two steps: (1) select one of the three low-level MoEs with probabilities $\lambda^x, \lambda^y, \lambda^{xy}$, and (2) apply random masking within the selected subset before feeding into the Transformer encoder.

It is straightforward to note that, using $p_\theta^{\text{HMoE}}$, maximizing $I_\theta(Z; XY)$ also maximizes a weighted combination of the MI terms, $\lambda^x \cdot I_\theta(Z_X; X) + \lambda^y \cdot I_\theta(Z_Y; Y) + \lambda^{xy} \cdot I_\theta(Z_{XY}; XY)$. In our experiments, we set $\lambda^x = \lambda^y = 0.5$, $\lambda^{xy} = 0$ to focus on improving the informativeness of the single-modality representations.

We then maximize $I_\theta(Z; XY)$ by reconstructing both modalities from $z \sim p_\theta^{\text{HMoE}}$ using the decoders, which yields the masked reconstruction objective:

$$\mathcal{J}_1(\theta, \phi) := \mathbb{E}_{p_D(xy)p_\theta^{\text{HMoE}}(z|xy)} \left[ \log q_\phi(xy \mid z) \right]. \tag{7}$$

However, it does not explicitly align representations across modalities, because the decoder can learn a many-to-one mapping that sends different inputs to the same target. Such ambiguity undermines the quality of the learned representations and prevents reliable retrieval.

To address this issue, we adopt the InfoNCE objective (Oord et al., 2018; Poole et al., 2019) as a variational lower bound of $I_\theta(Z; XY)$, which is defined below.

$$I_\theta(Z; XY) \geq \hat{I}_\theta^{NCE}(Z; XY) := \mathbb{E}_{\substack{\prod_{i=1}^K p_D(xy^{(i)}) \\ p_\theta^{\text{HMoE}}(z^{(i)}|xy^{(i)}) \\ p_\theta^{xy}(z_{xy}^{(i)}|xy^{(i)})}} \left[ \frac{1}{K} \sum_{i=1}^K \log \frac{e^{f\left(z^{(i)}, z_{xy}^{(i)}\right)/\tau}}{\frac{1}{K}\sum_{j=1}^K e^{f\left(z^{(i)}, z_{xy}^{(j)}\right)/\tau}} \right], \tag{8}$$

where $f$ computes the cosine similarity, $z_{xy}$ is the complete representation obtained by the full-submodality encoder $p_\theta^{xy}(z_{xy}|xy)$ [1], $K$ is the mini-batch size, and $\tau$ is the temperature. Thus, optimizing $\hat{I}_\theta^{NCE}$ not only increases MI but also aligns single-modality representations $z_x$ and $z_y$ with $z_{xy}$, aligning representations from different modalities.

Although $\hat{I}_\theta^{NCE}$ does not involve decoders for training, they are still needed for generation tasks such as mapping from PSMILES to property or vice versa. Therefore, we combine Equation 7 with Equation 8 to form the final objective:

$$\mathcal{J}_2(\theta, \phi) := \mathbb{E}_{p_D(xy)p_\theta^{\text{HMoE}}(z|xy)} \left[ \log q_\phi(xy \mid z) \right] + \beta \cdot \hat{I}_\theta^{NCE}(Z; XY). \tag{9}$$

This formulation balances reconstruction fidelity against cross-modal representation alignment through the trade-off factor $\beta$.

Beyond alignment, integrating the reconstruction objective (Equation 7) with the contrastive learning objective (Equation 8) can be viewed as a form of multi-task regularization (Caruana, 1997; Baxter, 2000; Argyriou et al., 2006), encouraging more robust representations that generalize across diverse downstream tasks. The overall training flow of MMPAE under the combined objective in Equation 9 is illustrated in Figure 2.

---

[1]$p_\theta^{xy}(z_{xy}|xy)$ is also one of the subset experts included in the MoME $p_\theta^{xy}(z \mid xy)$, because the complete-modal $xy$ itself belongs to the multimodal subset satisfying $xy \in P(xy) \setminus (P(x) \cup P(y))$.

## 4 EXPERIMENTS

To evaluate the versatility of our method, we conduct downstream experiments covering unimodal tasks, where only one modality is available during inference, and multimodal tasks, where both structural and property modalities are provided. Our evaluation consists of three stages: (1) large-scale benchmarking on polyOne, (2) real-world validation on POINT[2], and (3) analysis of model behavior via systematic comparison of MMPAE variants. Details of polyOne and implementation are provided in Appendix B.1 and B.2. A detailed analysis of the experimental results and underlying mechanisms is presented in Appendix D.

**MMPAE variants** For comprehensive evaluation, we consider three variants of our framework. (1) MMPAE optimizes the reconstruction objective in Equation 2 with uniform weights on all sub-modality experts. (2) MMPAE+HMoE extends MMPAE by applying the hierarchical mixture-of-experts objective in Equation 7 to increase the weights on unimodal experts, thereby enhancing the informativeness of single-modality representations. (3) MMPAE+InfoNCE, our final and recommended model, further incorporates the InfoNCE objective of Equation 9 to align representations across modalities while retaining the unimodal weighting of HMoE. Appendix K provides algorithmic descriptions of all variants together with corresponding schematic illustrations for clarity.

**Baseline methods** We compare MMPAE with both multimodal and unimodal baselines. As the multimodal baseline, we use SPMM across all downstream tasks because it jointly learns structure–property representations, and employs random token masking to ensure a fair comparison. For unimodal baselines, we adopt task-specific models: PolyBERT and TransPolymer for property prediction, and PolyTAO and Llamol for inverse design (see Appendix B.3 for details). To isolate the effect of multimodal design, we implement two unimodal variants that use the same backbone as MMPAE: the Property Transformer removes the PSMILES decoder, and the Inverse Transformer removes the property decoder. Unimodal models are excluded from cross-modal retrieval because they lack a shared latent space for modality alignment.

**Evaluation protocols** To assess representation robustness, we evaluate all models under two settings: (1) Complete Input, where all submodalities are fully observed, and (2) Missing Input, where structural tokens or property values are randomly masked (0–50%). Performance is measured using task-specific metrics. For property prediction, RMSE is reported against ground-truth values. For inverse design, we evaluate chemical correctness, structural similarity, and target property alignment via validity, Tanimoto similarity, and RMSE. For cross-modal retrieval, PSMILES and properties are retrieved by masking only the query input, and Top-K accuracy based on cosine similarity is reported. The task description and Further metric definitions are provided in Appendix B.4 and B.5.

### 4.1 POLYONE

We conduct large-scale evaluation on the polyOne dataset, which serves as the primary benchmark for assessing the core capabilities of MMPAE. Three representative downstream tasks are considered: property prediction, polymer inverse design, and cross-modal retrieval. The full quantitative results are provided in Appendix C.

#### 4.1.1 PROPERTY PREDICTION

Figure 3 reports RMSE under increasing levels of missing structural tokens. All methods deteriorates as masking increases, but MMPAE+InfoNCE achieve substantially lower errors across all missing ratio. This suggests that the combination of expert reweighting in HMoE and alignment via InfoNCE yields robust structural representations under unimodal inference. SPMM does not effectively utilize its additional modality and falls behind even unimodal baselines. Among unimodal baselines, Property Transformer shows the strongest robustness across all missing ratios, whereas PolyBERT and TransPolymer remain consistently weaker. To further support the statistical analysis, Appendix I reports property-wise standard deviations of prediction errors, and Appendix J provides property-wise scatter plots comparing ground-truth and predicted values for all 29 properties.

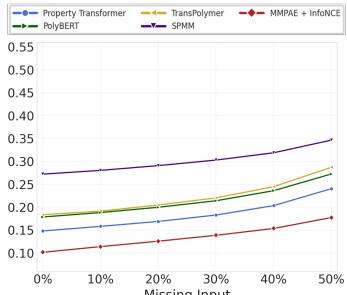

Figure 3: Property prediction results under missing inputs.

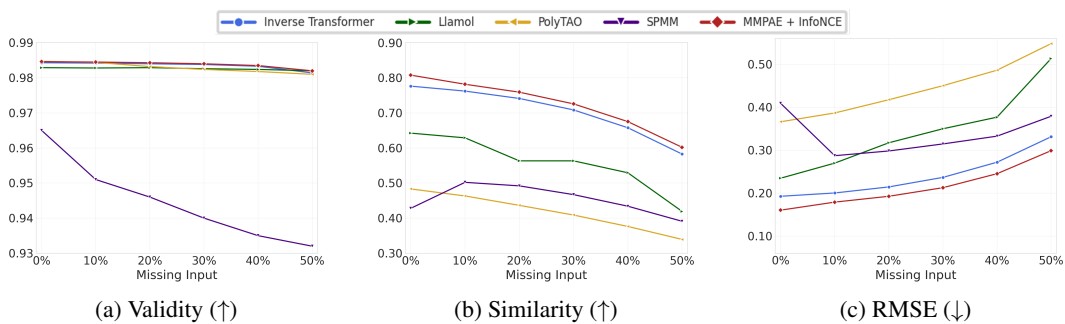

(a) Validity (↑)      (b) Similarity (↑)      (c) RMSE (↓)

Figure 4: Results in the inverse design task. The x-axis shows the fraction of missing input property values during evaluation, and the y-axis reports (a) Validity, (b) Similarity, and (c) RMSE.

### 4.1.2 POLYMER INVERSE DESIGN

Figure 4 reports inverse design performance under increasing missing input ratio. Validity remains high for most methods except SPMM, which deteriorates sharply as masking increases. Among unimodal baselines, the Inverse Transformer is the most competitive, whereas PolyTAO and Llamol show limited robustness and larger deviations from target properties. MMPAE+InfoNCE achieves highest performance across all masking ratios, suggesting that alignment improves the stability of representations under incomplete inputs. Qualitative examples are included in Appendix H.

### 4.1.3 CROSS-MODAL RETRIEVAL

Figure 5 shows Top-1 retrieval accuracy as query incompleteness increases, with Top-3 and Top-5 results in Section E.2. SPMM deteriorates rapidly, indicating that it fails to exploit complementary cues under partial observations. In contrast, MMPAE+InfoNCE consistently achieves the highest accuracy and exhibits the slowest degradation. This suggests that explicit contrastive alignment, reinforced by expert reweighting, facilitates stable cross-modal matching by preserving informative modality-specific signals under incomplete inputs.

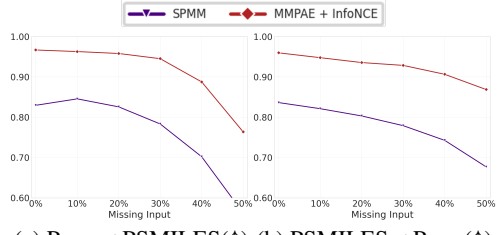

(a) Prop.→PSMILES(↑) (b) PSMILES→Prop.(↑)

Figure 5: Cross-modal retrieval accuracy (Top-1)

### 4.2 POINT$^2$

While polyOne provides large-scale benchmarking, its synthetic nature limits real-world applicability. We therefore evaluate on the Point$^2$ dataset (Xu et al., 2025), which contains experimentally collected polymer structures and properties. Point$^2$ is smaller in scale and exhibits naturally occurring missing values across property dimensions. These naturally occurring missing entries are explicitly treated as masked inputs in our submodality-masking protocol, enabling evaluation under realistic experimental conditions. Additional results on the OpenPoly dataset, another real-world benchmark, are provided in Appendix F.1.

### 4.2.1 PROPERTY PREDICTION

The overall trend under real-world conditions follows the earlier results: performance decreases as input incompleteness grows. MMPAE+InfoNCE maintains the lowest errors across all masking ratios, indicating robustness despite the smaller data scale and substantial missing values. Unlike the PolyOne setting, the Property Transformer exhibits inferior robustness compared to other unimodal baselines, implying that learning from a single modality becomes particularly ineffective when real datasets contain heterogeneous and partially observed inputs. polyBERT and TransPolymer show relatively stronger performance, which is likely attributable to their pretrained initializations rather than a clear architectural advantage. SPMM remains the weakest method overall.

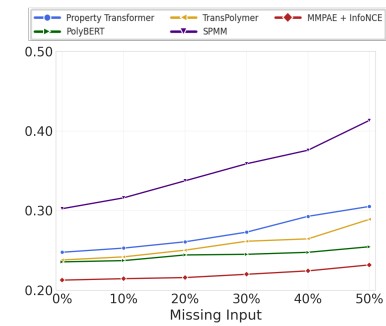

Figure 6: Property prediction results.

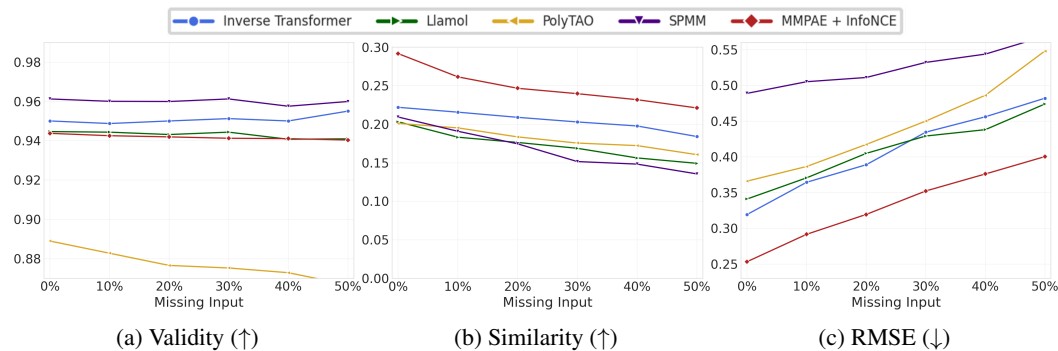

(a) Validity (↑)    (b) Similarity (↑)    (c) RMSE (↓)

Figure 7: Inverse design results on Point$^2$ under varying missing property ratios.

### 4.2.2 POLYMER INVERSE DESIGN

Figure 7 shows that MMPAE+InfoNCE achieves the best overall performance and maintains stable property alignment under severe missing inputs. The inherent missing values in Point$^2$ intensify the task difficulty, widening the gap between methods. SPMM attains high validity but lower similarity and RMSE, implying that chemical correctness alone is insufficient to ensure property fidelity.

### 4.2.3 CROSS-MODAL RETRIEVAL

Figure 8 illustrates the Top-1 retrieval accuracy on the Point$^2$. Consistent with earlier results, MMPAE+InfoNCE maintains the highest accuracy across all missing ratios and reliably surpasses SPMM, indicating stronger robustness to incomplete inputs. In addition, performance is consistently lower when properties are used as queries rather than PSMILES tokens. This gap likely arises because the missing-value pattern and masking protocol reduce the

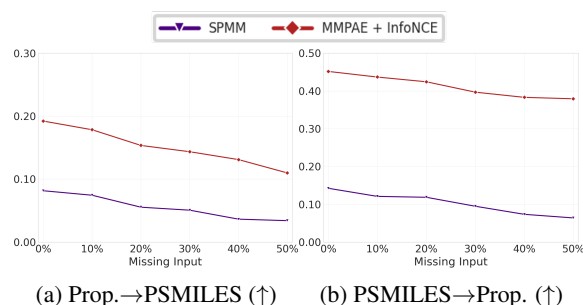

(a) Prop.→PSMILES (↑)    (b) PSMILES→Prop. (↑)

Figure 8: Cross-modal retrieval performance (Top-1)

availability of informative property signals, making cross-modal matching more challenging in the property-to-PSMILES direction.

### 4.3 ABLATION STUDY

We conduct a module-level ablation study to isolate the contributions of each component in MM-PAE. In addition to the main variants, we evaluate a configuration that disables HMoE while retaining InfoNCE, denoted as MMPAE+InfoNCE (w/o HMoE), to independently assess the alignment module. Additional ablations regarding InfoNCE hyperparameters are provided in Appendix G.

### 4.3.1 PROPERTY PREDICTION

As shown in Figure 9, MMPAE exhibits the lowest performance, indicating that uniform weighting of submodalities fails to capture modality-specific structure. Applying InfoNCE slows performance degradation, but the improvement remains limited because the alignment does not explicitly promote modality-specific representations required for unimodal inference. Introducing HMoE significantly improves performance across all missing ratios by reweighting unimodal experts and enhancing their modality-specific representations. Combining HMoE with InfoNCE further yields stable gains by aligning unimodal embeddings with their complete-submodality counterparts. The improvement stems from both higher mutual information and the joint regularization effects of contrastive alignment and reconstruction, which stabilize the latent space across masking ratios.

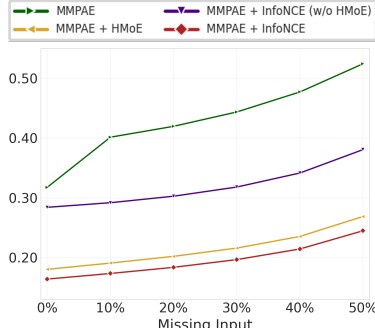

Figure 9: Property prediction results of MMPAE variants.

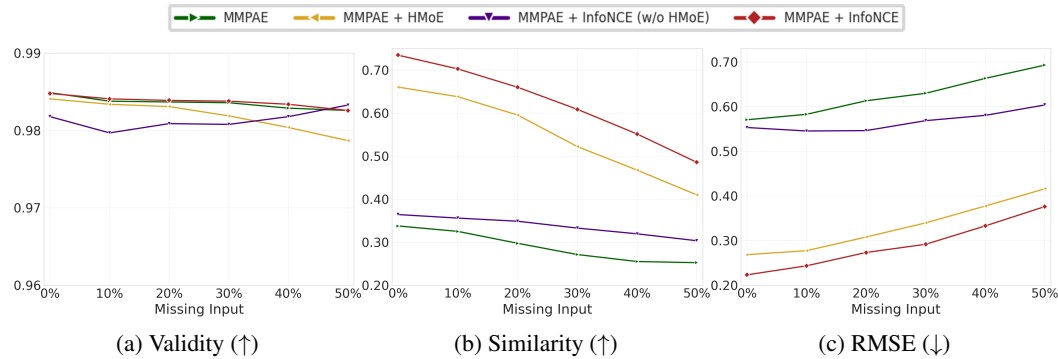

(a) Validity (↑)  (b) Similarity (↑)  (c) RMSE (↓)

Figure 10: Inverse design result of MMPAE variants. The x-axis shows the fraction of missing input property values during evaluation, and the y-axis reports (a) Validity, (b) Similarity, and (c) RMSE.

### 4.3.2 POLYMER INVERSE DESIGN

MMPAE performs considerably worse than all other methods in both similarity and RMSE despite its high validity, indicating that naive multimodal masking suppresses unimodal information and hinders the capture of property-specific signals. Applying only alignment (MMPAE+InfoNCE w/o HMoE) increases robustness but does not ensure conditional consistency, InfoNCE focuses on aligning global representations rather than preserving the property-conditioned structure needed for generation. In contrast, MMPAE+HMoE substantially improves performance by explicitly reweighting unimodal MI terms and preserving informative property representations. MMPAE+InfoNCE further enhances inference under incomplete inputs by aligning unimodal representations with their complete-submodality counterparts, enhancing conditional inference even under incomplete inputs.

### 4.3.3 CROSS-MODAL RETRIEVAL

Pure MMPAE and MMPAE+InfoNCE (w/o HMoE) is omitted due to extremely poor performance. Accuracy drops for all methods as missingness grows. Our final model, MMPAE+InfoNCE, delivers the highest overall accuracy, maintaining a clear margin over MMPAE+HMoE and showing especially strong gains at high missing rates, where explicit cross-modal alignment with the joint complete representation enables better use of complementary information. MMPAE+HMoE remains competitive under fully observed inputs but degrades sharply beyond a 0.2 masking rate, reflecting limited robustness without alignment.

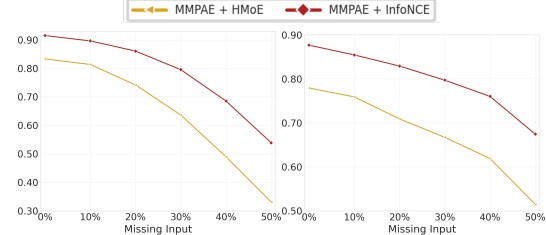

(a) Prop.→PSMILES (↑)   (b) PSMILES→Prop. (↑)

Figure 11: Cross-modal retrieval performance (Top-1)

## 5 CONCLUSION

This work introduces MMPAE, a multimodal representation framework that learns a shared latent space for polymer structures (PSMILES) and numerical properties while flexibly handling missing submodalities. By combining masked-reconstruction with hierarchical mixture-of-experts reweighting and an InfoNCE alignment objective, the model captures both unimodal informativeness and cross-modal consistency. Extensive experiments on large polymer datasets demonstrate strong performance across property prediction, inverse design, and cross-modal retrieval, particularly under incomplete inputs where conventional unimodal or purely generative approaches degrade sharply. These results highlight the importance of adaptive unimodal weighting and explicit cross-modal alignment as a unified foundation for robust polymer informatics.

## ETHICS STATEMENT

We have read the ICLR Code of Ethics. This work advances polymer informatics to accelerate the discovery of materials. Our study utilizes PolyOne dataset, a large and generally available resource created for polymer research. We acknowledge that this dataset, like other large data corpus, may contain inherent biases. Moreover, MMPAE enables property-conditioned polymer generation, which, while intended for positive scientific impact, could be misused for harmful applications. We emphasize the importance of transparency and community norms to mitigate dual-use risks. The ethical responsibility for preventing misuse rests with end users and deploying organizations, and requires adherence to community-established norms and guidelines.

## REPRODUCIBILITY STATEMENT

To ensure the reproducibility of our results, we provide the source code along with detailed descriptions of hyperparameters, data preprocessing, and additional implementation details in Section B. The PolyOne dataset used in our experiments is publicly available at Zenodo.

- Code: https://anonymous.4open.science/r/MMPAE-4F3C
- Dataset(PolyOne): https://zenodo.org/records/7766806

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

# Supplementary Material

## CONTENTS

## A  THEORETICAL RESULTS

### A.1  PROOF OF PROPOSITION 1

**Proposition 1.**  Given the MoE joint encoder $p_\theta$ defined as Equation 1,

$$I_\theta(Z; XY) \leq \frac{1}{2^{|X|+|Y|}} \sum_{S \in \mathcal{P}(XY)} I_\theta(Z_S; S).$$

*Proof.*  We first need to note that the marginal distribution of the joint representation $p_\theta^J(z)$ is a mixture of $p_\theta^s(z)$, where $p_\theta^s(z)$ is the marginal distribution of each subset-view representation as below.

$$p_\theta^J(z) = \int p_D(xy) p_\theta^J(z \mid xy) dxy = \int p_D(xy) \sum_{s \in \mathcal{P}(xy)} \frac{1}{2^{|X|+|Y|}} \cdot p_\theta^s(z|s) \, dxy$$

$$= \sum_{s \in \mathcal{P}(xy)} \frac{1}{2^{|X|+|Y|}} \int p_D(s) p_\theta^s(z|s) ds$$

$$= \sum_{s \in \mathcal{P}(xy)} \frac{1}{2^{|X|+|Y|}} \cdot p_\theta^s(z), \tag{10}$$

where $p_D(s) = \int p_D(xy) \, d(xy \backslash s)$. Thus,

$$I_\theta(Z; XY) = \mathbb{E}_{p_D(xy)} \left[ D_{KL} \left[ p_\theta^J(z \mid xy) || p_\theta^J(z) \right] \right] \tag{11}$$

$$= \mathbb{E}_{p_D(xy)} \left[ D_{KL} \left[ \sum_{s \in \mathcal{P}(xy)} \frac{1}{2^{|X|+|Y|}} \cdot p_\theta^s(z|s) || \sum_{s \in \mathcal{P}(xy)} \frac{1}{2^{|X|+|Y|}} \cdot p_\theta^s(z) \right] \right] \tag{12}$$

$$\leq \mathbb{E}_{p_D(xy)} \left[ \sum_{s \in \mathcal{P}(xy)} \frac{1}{2^{|X|+|Y|}} \cdot D_{KL} \left[ p_\theta^s(z|s) || p_\theta^s(z) \right] \right] \tag{13}$$

$$= \sum_{s \in \mathcal{P}(xy)} \frac{1}{2^{|X|+|Y|}} \cdot \mathbb{E}_{p_D(s)} \left[ D_{KL} \left[ p_\theta^s(z|s) || p_\theta^s(z) \right] \right]$$

$$= \sum_{S \in \mathcal{P}(XY)} \frac{1}{2^{|X|+|Y|}} \cdot I_\theta(Z_S; V_S).$$

Equation 12 holds because the latter term in KL in Equation 11 can be decomposed into $p_\theta^J(z) = \sum_{s \in \mathcal{P}(xy)} \frac{1}{2^{|X|+|Y|}} p_\theta^s(z)$ as in Equation 10. Lastly, the inequality in Equation 13 holds due to the convexity of KL divergence. □

## A.2 CONNECTION TO MULTIMODAL VAES

Multimodal VAEs (Wu & Goodman, 2018; Shi et al., 2019; Sutter et al., 2021; Hwang et al., 2021) learn joint representations from multiple modalities using late-fusion strategies such as Mixture of Experts (MoE), Product of Experts (PoE), and Mixture-of-Product-of-Experts (MoPoE). Remarkably, MoPoE builds experts over all possible subsets of modalities and aggregate their outputs to capture both shared and modality-specific information. In contrast, our joint encoder $p_\theta^J$ and $p_\theta^{\text{HMoE}}$ adopts a Transformer-based early-fusion variant of MoE, directly operating on the concatenated embeddings of all submodalities while retaining the expert decomposition. Because it contains experts of the same functional form, our encoder naturally subsumes the Transformer implementations of MVAE methods as special cases. Specifically, by substituting experts of complete intra-modality subsets for the low-level MoEs (e.g., $p_\theta^x(z \mid x) = p_{s_x=x}(z|s_x = s)$) and adjusting the mixture weights $\lambda^x$, $\lambda^y$, and $\lambda^{xy}$ in $p_\theta^{\text{HMoE}}$, we can recover several well-known MVAE configurations:

1. $\lambda^x = \lambda^y = 0.5, \lambda^{xy} = 0$: MMVAE (Shi et al., 2019).
2. $\lambda^x = \lambda^y = 0, \lambda^{xy} = 1$: MVAE (Wu & Goodman, 2018), MVTCAE (Hwang et al., 2021).
3. $\lambda^x = \lambda^y = \lambda^{xy} = \frac{1}{3}$: MoPoE-VAE (Sutter et al., 2021).

While these MVAE approaches can handle missing modalities, they cannot address cases of missing submodalities within a modality—for example, generating PSMILES sequences from only a subset of property values or inferring property values from partial PSMILES tokens. Our early-fusion encoder extends beyond standard MVAE capabilities by explicitly modeling and aggregating such fine-grained submodality combinations.

# B DETAILS IN EXPERIMENTS

## B.1 DATASET

We evaluate our method on the PolyOne dataset (Kuenneth & Ramprasad, 2022) introduced in Poly-BERT. PolyOne is one of the largest publicly available polymer datasets, comprising approximately 100 million structure–property pairs. Although PolyOne is large in scale, we empirically observed that increasing the training size beyond 5 million samples did not yield further performance improvements across baselines and tasks. We therefore fix the training set at 5 million instances, with 100 thousand each for validation and testing, to ensure both efficiency and comparability. Each entry contains two modalities: a PSMILES sequence and 29 property values. PSMILES strings are tokenized into fixed-length sequences of 160 tokens using the pretrained PolyBERT tokenizer. All properties are normalized per dimension using z-scores to ensure comparability across units.

## B.2 IMPLEMENTATION DETAIL

Our Multimodal Masked Polymer Autoencoder (MMPAE) consists of three parts: Transformer encoder, Autoregressive PSMILES decoder, and property decoder. All the hyperparameters we used are below.

| Hyperparameter | Value |
| --- | --- |
| Number of parameters | 0.355B |
| Number of training epochs | 200 |
| Batch size | 512 |
| Learning rate | 1e-4 |
| Embedding dimensions | 1,024 |
| Representation dimensions | 1,024 |
| Feedforward layer dimensions | 2,048 |
| Use bias in Feedforward layers | True |
| Number of attention heads | 16 |
| Encoder depth | 24 |
| Transformer decoder depth | 12 |
| MLP decoder depth | 1 |
| Activation function | GELU |
| Dropout rate | 0.0 |
| Normalization | pre-norm |
| $\beta$ | 1,000 |
| $\tau$ | 0.2 |

Table 1: Hyperparameters of MMPAE.

### B.3 Baselines Detail

**polyBERT**    polyBERT is a BERT-based encoder pretrained on large PSMILES corpora and fine-tuned for regression tasks.

**TransPolymer**    TransPolymer extends the masked modeling paradigm with task-adaptive pretraining and achieves improved property prediction on curated datasets. We use their public checkpoints and fine-tune under our protocol.

**Property Transformer**    To isolate the effect of multimodal fusion, we design a unimodal variant that shares the encoder backbone of MMPAE but excludes the PSMILES decoder, yielding a Transformer specialized for property prediction. This controlled design enables fair comparison with MMPAE under identical architectural capacity.

**PolyTAO**    PolyTAO generates PSMILES sequences autoregressively, conditioned on discretized property tokens. We follow the released implementation and restrict conditioning to numerical properties for consistency with our setup.

**Llamol**    Llamol supports fragment-wise and continuous generation. We adopt the continuous generation mode and condition only on numerical properties to align with our inverse design task.

**Inverse Transformer**    As a controlled unimodal baseline, we implement an Inverse Transformer that shares the backbone of MMPAE but excludes the property decoder, enabling property-conditioned generation from PSMILES inputs alone.

**SPMM**    SPMM learns joint structural–property representations via random token masking. As its original implementation does not assume missing inputs, we apply random masking to PSMILES tokens during training to ensure a fair comparison under missing-token conditions.

### B.4 Task Description

**Property Prediction**    Property prediction is a fundamental benchmark in polymer informatics, formulated as a regression task where the model maps a structural representation to continuous property values. This task assesses how effectively the learned representation captures complex structure–property relationships.

**Polymer Inverse Design**    Polymer inverse design is formulated as a conditional sequence generation task where the model generates chemically valid PSMILES sequences that conditioned on target properties. Unlike forward prediction, this setting directly tests whether learned representations capture the underlying structure–property relationships. The challenge lies in the large chemical search space and the requirement to satisfy multiple interdependent properties simultaneously.

**Cross-modal Retrieval**    Cross-modal retrieval evaluates whether structural and property representations are aligned in a shared latent space. Given a query in one modality, such as a PSMILES sequence or a property vector, the model retrieves the most relevant candidates from the other modality. This task reflects practical scenarios: researchers often pre-select feasible candidates and must identify the one that best satisfies desired property constraints. Because polymer datasets frequently contain missing property values (Kuenneth et al., 2021; Malashin et al., 2023; Yuan et al., 2021; Parvez & Mehedi, 2025), learning representations that remain reliable under incomplete inputs is essential. Direct nearest-neighbor search in raw property space becomes unreliable when key dimensions are missing, whereas encoding both modalities into a unified latent space enables more robust retrieval under such conditions.

## B.5 METRICS

We evaluate the generated polymers using four widely adopted metrics: Validity, Tanimoto similarity, RMSE, and $R^2$-Score.

- *Validity* measures the proportion of generated PSMILES that correspond to chemically valid and syntactically parsable polymer structures, as determined by RDKit Landrum et al. (2006), a cheminformatics toolkits.

- *Tanimoto similarity* quantifies structural similarity between generated and reference polymers, typically computed using fingerprint-based representations (e.g., Morgan fingerprints).

- *RMSE* and $R^2$-*Score* evaluate how well the predicted properties of the generated polymers match the conditioning targets.

To obtain these property estimates, we employ a Property Transformer trained with complete submodalities as part of our baseline models during the evaluation in inverse design tasks (Section 4.1.2), following the standard evaluation protocol (Gómez-Bombarelli et al., 2018; Yue et al., 2025).

# C QUANTITATIVE RESULTS

This section provides the complete numerical results for all downstream tasks evaluated in the main manuscript, including property prediction, polymer inverse design, and cross-modal retrieval. The main text highlights high-level comparisons and robustness patterns.

## C.1 PROPERTY PREDICTION

Table 2: Property prediction performance under different missing input ratios. RMSE ($\downarrow$) and $R^2$ ($\uparrow$) are reported for variants of MMPAE and all methods.

| Method | Metric | Missing Input Ratio | | | | | |
|---|---|---|---|---|---|---|---|
| | | 0.0 | 0.1 | 0.2 | 0.3 | 0.4 | 0.5 |
| **MMPAE + InfoNCE** | RMSE | **0.1014** | **0.1139** | **0.1257** | **0.1386** | **0.1535** | **0.1773** |
| | $R^2$ | **0.9893** | **0.9865** | **0.9836** | **0.9800** | **0.9754** | **0.9672** |
| **MMPAE + HMoE** | RMSE | 0.1177 | 0.1308 | 0.1414 | 0.1517 | 0.1695 | 0.1974 |
| | $R^2$ | 0.9869 | 0.9839 | 0.9808 | 0.9769 | 0.9714 | 0.9601 |
| **MMPAE** | RMSE | 0.5046 | 0.5178 | 0.5343 | 0.5552 | 0.5841 | 0.6212 |
| | $R^2$ | 0.7404 | 0.7267 | 0.7090 | 0.6857 | 0.6521 | 0.6062 |
| **SPMM** | RMSE | 0.2722 | 0.2804 | 0.2908 | 0.3029 | 0.3188 | 0.3465 |
| | $R^2$ | 0.9133 | 0.9096 | 0.9043 | 0.8978 | 0.8888 | 0.8707 |
| **Property Transformer** | RMSE | 0.1481 | 0.1581 | 0.1689 | 0.1828 | 0.2034 | 0.2404 |
| | $R^2$ | 0.9769 | 0.9738 | 0.9701 | 0.9651 | 0.9569 | 0.9400 |
| **polyBERT** | RMSE | 0.1786 | 0.1883 | 0.1998 | 0.2140 | 0.2360 | 0.2725 |
| | $R^2$ | 0.9669 | 0.9631 | 0.9586 | 0.9525 | 0.9423 | 0.9228 |
| **TransPolymer** | RMSE | 0.1836 | 0.1914 | 0.2047 | 0.2203 | 0.2448 | 0.2871 |
| | $R^2$ | 0.9642 | 0.9582 | 0.9491 | 0.9380 | 0.9318 | 0.9157 |

## C.2 POLYMER INVERSE DESIGN

Table 3: Inverse design performance under different missing-input ratios. Validity (↑), Tanimoto similarity (↑), RMSE (↓), and $R^2$ (↑) are reported for variants of MMPAE and all methods.

| Method | Metric | Missing Input Ratio | | | | | |
|---|---|---|---|---|---|---|---|
| | | 0.0 | 0.1 | 0.2 | 0.3 | 0.4 | 0.5 |
| **MMPAE + InfoNCE** | Validity | 0.9846 | **0.9845** | 0.9840 | 0.9840 | **0.9835** | 0.9820 |
| | Tanimoto Sim. | **0.8077** | **0.7817** | **0.7591** | **0.7257** | **0.6752** | **0.6019** |
| | RMSE | **0.1603** | **0.1790** | **0.1926** | **0.2126** | **0.2452** | **0.2988** |
| | $R^2$ | **0.9699** | **0.9662** | **0.9609** | **0.9528** | **0.9372** | **0.9067** |
| **MMPAE + HMoE** | Validity | 0.9839 | 0.9838 | 0.9835 | 0.9821 | 0.9805 | 0.9781 |
| | Tanimoto Sim. | 0.7335 | 0.7172 | 0.6944 | 0.6392 | 0.5913 | 0.5262 |
| | RMSE | 0.2055 | 0.2128 | 0.2273 | 0.2605 | 0.2897 | 0.3386 |
| | $R^2$ | 0.9555 | 0.9526 | 0.9458 | 0.9290 | 0.9122 | 0.8800 |
| **MMPAE** | Validity | **0.9847** | 0.9842 | **0.9841** | 0.9838 | 0.9830 | 0.9820 |
| | Tanimoto Sim. | 0.4109 | 0.4043 | 0.3966 | 0.3883 | 0.3789 | 0.3687 |
| | RMSE | 0.5074 | 0.5182 | 0.5323 | 0.5506 | 0.5755 | 0.6155 |
| | $R^2$ | 0.6826 | 0.6666 | 0.6462 | 0.6188 | 0.5813 | 0.5155 |
| **SPMM** | Validity | 0.9652 | 0.9505 | 0.9462 | 0.9370 | 0.9352 | 0.9320 |
| | Tanimoto Sim. | 0.4280 | 0.5020 | 0.4920 | 0.4670 | 0.4340 | 0.3910 |
| | RMSE | 0.4123 | 0.2921 | 0.3022 | 0.3180 | 0.3366 | 0.3809 |
| | $R^2$ | 0.8373 | 0.9280 | 0.9172 | 0.9014 | 0.8862 | 0.8438 |
| **Inverse Transformer** | Validity | 0.9840 | 0.9840 | 0.9838 | 0.9836 | 0.9830 | 0.9809 |
| | Tanimoto Sim. | 0.7763 | 0.7412 | 0.7085 | 0.7085 | 0.6577 | 0.5826 |
| | RMSE | 0.2026 | 0.2097 | 0.2186 | 0.2378 | 0.2684 | 0.3241 |
| | $R^2$ | 0.9565 | 0.9535 | 0.9494 | 0.9403 | 0.9240 | 0.8895 |
| **PolyTAO** | Validity | **0.9847** | 0.9844 | 0.9832 | **0.9844** | 0.9808 | 0.9810 |
| | Tanimoto Sim. | 0.4832 | 0.4632 | 0.4364 | 0.4088 | 0.3763 | 0.3393 |
| | RMSE | 0.3661 | 0.3866 | 0.4174 | 0.4501 | 0.4862 | 0.5481 |
| | $R^2$ | 0.8600 | 0.8421 | 0.8182 | 0.7868 | 0.7443 | 0.6761 |
| **Llamol** | Validity | 0.9840 | 0.9843 | **0.9841** | 0.9832 | 0.9824 | **0.9823** |
| | Tanimoto Sim. | 0.6376 | 0.6251 | 0.5950 | 0.5619 | 0.5285 | 0.4197 |
| | RMSE | 0.2841 | 0.3016 | 0.3328 | 0.3427 | 0.3543 | 0.4526 |
| | $R^2$ | 0.9210 | 0.9108 | 0.8915 | 0.8836 | 0.8758 | 0.7947 |

## C.3 CROSS-MODAL RETRIEVAL

Table 4: Cross-modal retrieval performance of all models under different missing-input ratios. We report Recall@1/3/5 for both PSMILES→Property and Property→PSMILES directions.

| Method | Direction | Metric | Missing Input Ratio | | | | | |
|---|---|---|---|---|---|---|---|---|
| | | | 0.0 | 0.1 | 0.2 | 0.3 | 0.4 | 0.5 |
| **MMPAE + InfoNCE** | PSMILES → Prop. | R@1 | **0.9598** | **0.9448** | **0.9457** | **0.9287** | **0.9066** | **0.8688** |
| | | R@3 | **0.9958** | **0.9897** | **0.9849** | **0.9765** | **0.9638** | **0.9388** |
| | | R@5 | **0.9979** | **0.9940** | **0.9903** | **0.9842** | **0.9743** | **0.9534** |
| | Prop. → PSMILES | R@1 | **0.9669** | **0.9630** | **0.9682** | **0.9453** | **0.8877** | **0.7638** |
| | | R@3 | **0.9963** | **0.9953** | **0.9933** | **0.9818** | **0.9448** | **0.8490** |
| | | R@5 | **0.9981** | **0.9975** | **0.9959** | **0.9873** | **0.9568** | **0.8727** |
| **MMPAE + HMoE** | PSMILES → Prop. | R@1 | 0.9325 | 0.9194 | 0.8955 | 0.8687 | 0.8354 | 0.7793 |
| | | R@3 | 0.9645 | 0.9599 | 0.9445 | 0.9270 | 0.9039 | 0.8623 |
| | | R@5 | 0.9863 | 0.9643 | 0.9517 | 0.9369 | 0.9173 | 0.8807 |
| | Prop. → PSMILES | R@1 | 0.9437 | 0.9390 | 0.9080 | 0.8451 | 0.7509 | 0.6148 |
| | | R@3 | 0.9765 | 0.9710 | 0.9555 | 0.9160 | 0.8489 | 0.6542 |
| | | R@5 | 0.9876 | 0.9737 | 0.9617 | 0.9288 | 0.8726 | 0.7435 |
| **MMPAE** | PSMILES → Prop. | R@1 | 0.0502 | 0.0120 | 0.0099 | 0.0081 | 0.0060 | 0.0041 |
| | | R@3 | 0.0930 | 0.0209 | 0.0176 | 0.0143 | 0.0107 | 0.0075 |
| | | R@5 | 0.1196 | 0.0270 | 0.0228 | 0.0184 | 0.0139 | 0.0099 |
| | Prop. → PSMILES | R@1 | 0.0730 | 0.0141 | 0.0116 | 0.0091 | 0.0068 | 0.0047 |
| | | R@3 | 0.1286 | 0.0244 | 0.0204 | 0.0161 | 0.0121 | 0.0087 |
| | | R@5 | 0.1622 | 0.0311 | 0.0263 | 0.0207 | 0.0159 | 0.0114 |
| **SPMM** | PSMILES → Prop. | R@1 | 0.8364 | 0.8213 | 0.8033 | 0.7792 | 0.7428 | 0.6770 |
| | | R@3 | 0.9627 | 0.9557 | 0.9469 | 0.9334 | 0.9104 | 0.8600 |
| | | R@5 | 0.9804 | 0.9762 | 0.9707 | 0.9619 | 0.9453 | 0.9053 |
| | Prop. → PSMILES | R@1 | 0.8297 | 0.8457 | 0.8258 | 0.7834 | 0.7021 | 0.5625 |
| | | R@3 | 0.9583 | 0.9656 | 0.9583 | 0.9376 | 0.8855 | 0.7680 |
| | | R@5 | 0.9784 | 0.9826 | 0.9788 | 0.9659 | 0.9289 | 0.8316 |

# D    DETAILED ANALYSIS OF EXPERIMENTAL RESULTS

This section provides an in-depth analysis of the experimental results, elucidating the underlying mechanisms driving MMPAE's robustness and the synergistic roles of its architectural components.

## D.1    PROPERTY PREDICTION

As shown in Figure 3 and Figure 6, across both the PolyOne and Point$^2$ datasets, MMPAE+InfoNCE consistently achieves superior performance. While predictive accuracy naturally degrades for all methods as input incompleteness increases, our model demonstrates significantly higher robustness with minimal performance decay. The core driver of this resilience is the explicit promotion of unimodal informativeness via the Hierarchical Mixture-of-Experts (HMoE). By upweighting unimodal experts, HMoE compels the model to extract meaningful features solely from partial inputs. Furthermore, the InfoNCE objective aligns these unimodal representations with their complete sub-modality counterparts. This alignment ensures that latent representations inferred from partial inputs remain sufficiently informative, effectively serving as a form of multi-task regularization alongside the reconstruction objective.

These advantages are particularly pronounced on the real-world Point$^2$ dataset. While the Property Transformer, a unimodal baseline sharing the MMPAE backbone, remains competitive on the large-scale PolyOne dataset, it lags significantly behind MMPAE+InfoNCE in the data-scarce and missing-value conditions of Point$^2$. This disparity suggests that unidirectional mapping (i.e., PSMILES $\rightarrow$ Property) is insufficient for capturing robust structural information under data constraints. In contrast, MMPAE+InfoNCE leverages a rich, shared latent space constructed through multimodal learning, providing empirical evidence of its robustness against both data scarcity and input incompleteness.

## D.2    POLYMER INVERSE DESIGN

Polymer inverse design effectively functions as a conditional generation task where the model must infer optimal structures from target properties. As input incompleteness increases, the available conditional information naturally diminishes, complicating precise mapping. MMPAE+InfoNCE counteracts this degradation by explicitly preserving the mutual information of partial inputs.

By employing HMoE, the model prevents the latent space from degenerating into an uninformative region during sparse input conditions. This ensures that the latent vector remains within a chemically valid generative manifold, allowing the decoder to reconstruct coherent chemical syntax even from weak signals. Simultaneously, the InfoNCE objective actively constrains the search space by aligning property representations with their structural counterparts. This effectively regularizes the inherent one-to-many mapping problem, enabling the model to approximate the target conditional distribution more accurately than baselines, even under severe information constraints.

A joint analysis of Validity against Similarity and RMSE in Figure 4 and Figure 7 reveals a critical distinction between syntactic correctness and semantic alignment. Regarding validity results across the PolyOne and Point$^2$ datasets, it is notable that most baselines maintain high validity even under significant input incompleteness. This suggests that these models have successfully internalized chemical syntax rules, such as valency and ring closure, regardless of conditioning sparsity. However, the semantic counterparts, Similarity and RMSE, present a conflicting narrative. For baseline models, high validity is accompanied by a sharp decline in structural similarity and a rapid increase in deviation from target properties (RMSE). This disparity indicates that while baselines can generate chemically plausible structures, they fail to reflect the target properties. In essence, the partial conditioning signal in baselines is insufficient to guide the latent vector to the correct functional region, resulting in the generation of valid but generic polymers that ignore target constraints.

In contrast, MMPAE+InfoNCE demonstrates a superior capability to couple syntax with semantics. It maintains high validity while achieving the lowest RMSE and highest relative similarity. This confirms that our model does not merely generate random valid strings but successfully leverages the aligned latent space to produce candidates that are both chemically feasible and functionally aligned with the target specifications.

## D.3    CROSS-MODAL RETRIEVAL

Experimental results in Figure 5 and Figure 8 reveal a consistent performance asymmetry between the two retrieval directions: retrieval from PSMILES to Properties yields higher accuracy than re-

trieval from Properties to PSMILES. This disparity is rooted in the intrinsic causality of polymer science. The mapping from structure to property is generally deterministic as a single polymer structure dictates a specific set of physicochemical properties. Conversely, the mapping from property to structure is an ill-posed, one-to-many problem, as multiple distinct structural analogs can exhibit identical physicochemical profiles. Consequently, retrieving a unique structural identifier solely from property values is more challenging. Despite this inherent difficulty, MMPAE achieves significantly higher Top-1 accuracy compared to SPMM, indicating that our model effectively clusters functionally similar structures within the latent space, thereby resolving the ambiguity of the inverse mapping more successfully.

The degradation patterns under missing input conditions highlight the specific efficacy of MMPAE's alignment strategy compared to SPMM. While SPMM also incorporates cross-modal alignment mechanisms, it exhibits a rapid performance collapse as query incompleteness increases. This suggests that its alignment objective may not sufficiently enforce latent invariance against severe input sparsity. In contrast, MMPAE+InfoNCE maintains robust retrieval accuracy by leveraging a more explicit anchor. Specifically, our InfoNCE objective forces the latent representation of masked submodalities to geometrically converge toward that of the complete submodality representation. This ensures that even a fragmented query is mapped to the correct semantic neighborhood defined by the full information content. Consequently, the partial representation effectively acts as a robust proxy for the complete input, enabling precise cross-modal retrieval even when the query is heavily compromised.

Superior retrieval performance serves as a direct proxy for evaluating the quality of the shared latent space. In multimodal representation learning, effective retrieval requires minimizing the geometric distance between a structure and its corresponding property vector. The fact that MMPAE outperforms baselines across all missing ratios confirms that the HMoE encoder preserves unimodal informativeness, while the InfoNCE loss ensures cohesive integration. This results in a highly structured latent manifold where structural and property information are effectively aligned, facilitating accurate cross-modal matching even under severe data sparsity.

## D.4 ABLATION STUDY

The inferior performance of the vanilla MMPAE baseline highlights the limitations of uniform submodality masking. Incorporating a naive masking with reconstruction objective inherently biases the model toward the joint modality, causing it to neglect unimodal dependencies. The introduction of HMoE addresses this by explicitly sampling unimodal subsets during training. This mechanism compels the model to learn independently informative representations from single modalities, ensuring robust feature extraction even when complementary information is entirely absent. Empirically, this is evidenced by the MMPAE+HMoE maintaining high accuracy in property prediction and generative consistency in inverse design, contrasting sharply with the severe degradation observed in the vanilla baseline. This confirms that the explicit unimodal exposure provided by HMoE successfully instills the necessary representational independence required for robust inference.

The results of the MMPAE + InfoNCE (w/o HMoE) variant offer a critical insight into the nature of contrastive alignment. Under standard random masking, encoder inputs are predominantly multimodal subsets containing partial information from both structure and property views. Consequently, the InfoNCE objective primarily learns to align these partial joint representations with the complete joint representation. The model thus learns to rely on inter-modal shortcuts present in the input to minimize contrastive loss, rather than learning to map a single modality to the global context. This reliance explains the variant's failure in downstream unimodal tasks (Figure 9 and Figure 10): it overfits to the presence of joint information. Thus, HMoE is essential not just for feature extraction but for correcting the alignment distribution, ensuring that the InfoNCE objective explicitly trains the model to align unimodal views with the global context.

The ablation study confirms that HMoE and InfoNCE play distinct yet complementary roles. HMoE ensures coverage by forcing the model to handle disjoint unimodal inputs, while InfoNCE ensures consistency by mapping those inputs to a unified latent manifold. The superior performance of the MMPA+InfoNCE validates that both components are indispensable, as HMoE prevents shortcut learning through joint cues and InfoNCE establishes a cohesive structure-property space.

# E ADDITIONAL EXPERIMENTS

## E.1 COMPLETE SUBMODALITY EXPERIMENTS

We additionally train all unimodal baseline models under the complete submodalities setting, where full information of inputs is always provided. Although such fully specified conditions are rarely encountered in practice, particularly for polymer properties, this experiment serves two complementary purposes compared to the masked submodalities setting. First, it establishes the upper-bound performance achievable when no input information is missing. Second, it enables a clearer evaluation of how performance degrades as missing information is introduced, thereby quantifying the robustness of each method to incomplete inputs. For this analysis, we restrict the complete submodalities experiments to unimodal baselines, while including the variants of MMPAE trained under the masked submodalities setting as additional points of comparison.

**Property prediction**

Figure 12 reports RMSE and $R^2$ Score across increasing levels of missing input. All unimodal baselines outperform our best masked-submodality model under the no missing input condition. However, this advantage diminishes rapidly once even a small fraction of inputs is absent; for example, unimodal baselines trained exclusively on complete submodalities exhibit an RMSE increase of more than 0.3 with only 10% missing inputs. This vulnerability arises from a distributional shift: models trained solely on fully specified inputs fail to learn representations that remain robust when confronted with incomplete information.

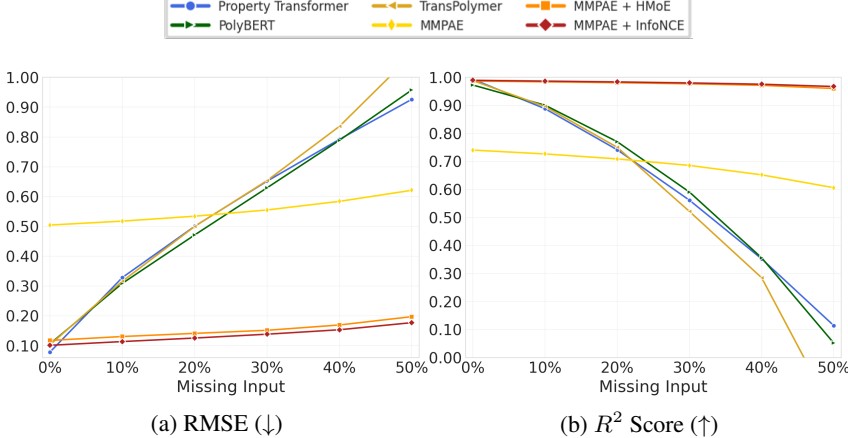

(a) RMSE (↓)     (b) $R^2$ Score (↑)

Figure 12: Property prediction result on complete submodality scenario. The x-axis shows the fraction of missing PSMILES tokens during evaluation, and the y-axis reports RMSE (↓).

**Polymer inverse design**

Figure 13 reports Validity, Similarity, RMSE, and $R^2$ Score across increasing masking ratios. All unimodal baselines can generate valid polymer structures even when parts of the input are missing. This robustness stems from their powerful transformer decoders, which are capable of producing chemically valid sequences even when masked submodalities limit the extraction of informative features. However, this phenomenon is reflected in the similarity, RMSE, and $R^2$ Score metrics: although unimodal models maintain high validity across all missing levels, their alignment between structural outputs and property conditions deteriorates sharply.

Notably, MMPAE+InfoNCE, trained under the masked-submodality setting, matches the upper-bound performance of the Inverse Transformer at 0% missing input. This finding demonstrates that the incorporation of the contrastive loss enables effective integration of information across modalities, leading to synergistic gains in structure–property alignment.

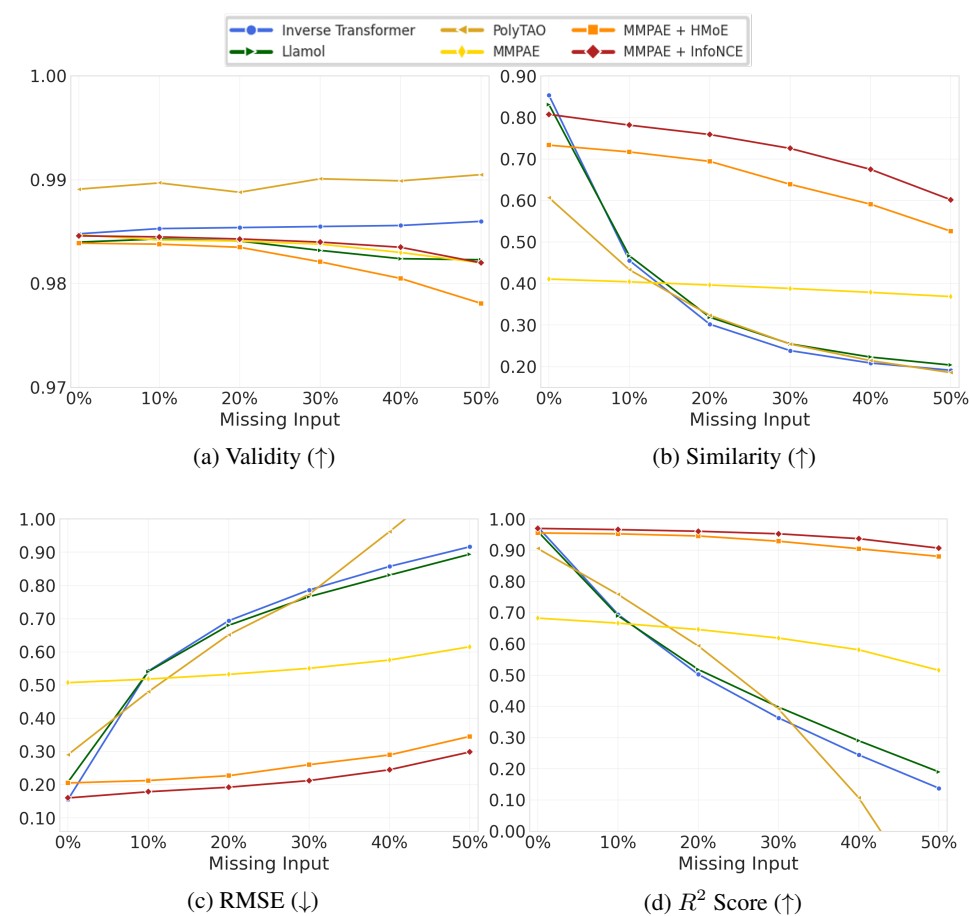

Figure 13: Inverse design result on complete submodality scenario. The x-axis shows the fraction of missing input property values during evaluation, and the y-axis reports (a) Validity, (b) Similarity, and (c) RMSE.

## E.2 ADDITIONAL RESULTS IN CROSS-MODAL RETRIEVAL

Similar to the Top-1 cross-modal retrieval results, MMPAE+InfoNCE consistently outperforms all baselines at Top-3 and Top-5, with the performance gap widening as the proportion of missing input increases.

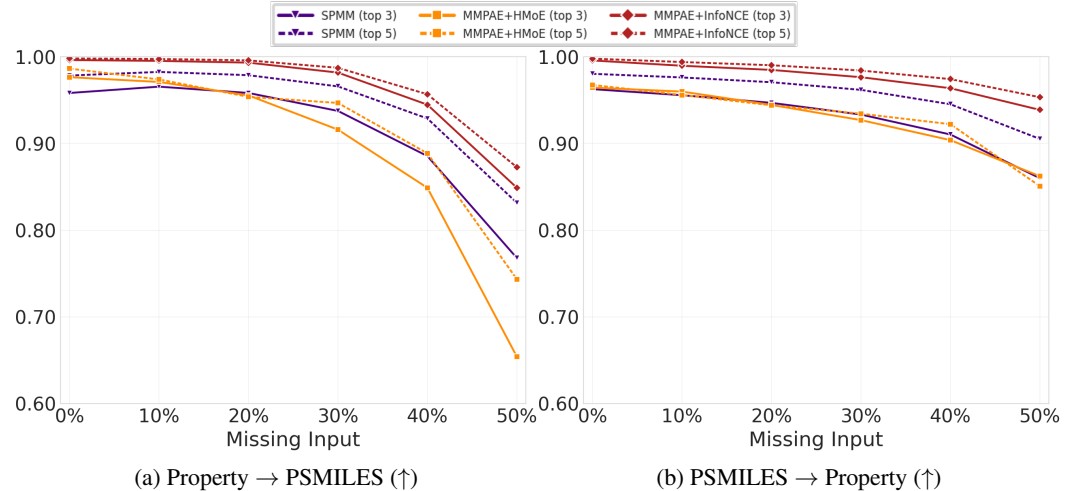

(a) Property → PSMILES (↑)  (b) PSMILES → Property (↑)

Figure 14: Additional retrieval result

## E.3 BRICS FRAGMENT MAKSING FOR PROPERTY PREDICTION

Our masking protocol independently drops each submodality, providing a controlled way to evaluate robustness under token- or property-level corruption. While this protocol is useful for controlled comparisons, it does not fully reflect practical settings in polymer informatics, where structural information is often missing at the level of entire chemical fragments rather than individual tokens. To better examine this setting, we extend the masking granularity from tokens to fragments in the property prediction task. To construct fragment masks, we perform BRICS decomposition on each PSMILES sequence and mask all tokens corresponding to a randomly selected fragment. Importantly, fragment-level masking is applied only during evaluation, without any additional training or fine-tuning.

As shown in Table 5, MMPAE+InfoNCE achieves the strongest performance under this more structured masking condition, outperforming both unimodal baselines and the multimodal model. At the same time, the other baselines also show reasonably robust performance. This is likely because the masking protocol models all combinations of PSMILES submodalities, enabling the models to retain useful information even when chemically

| Method | RMSE | $R^2$ |
|---|---|---|
| Property Transformer | 0.1602 | 0.9661 |
| PolyBert | 0.1987 | 0.9584 |
| TransPolymer | 0.2034 | 0.9537 |
| SPMM | 0.2974 | 0.9018 |
| **MMPAE+InfoNCE** | **0.1357** | **0.9805** |

Table 5: Property-prediction performance under BRICS fragment masking. Lower RMSE and higher $R^2$ indicate better performance.

meaningful fragments are removed. These findings confirm that our token-level masking protocol remains valid when evaluated under more realistic fragment-level missing settings.

## F ADDITIONAL REAL-WORLD EXPERIMENT

We further evaluated the generalizability of MMPAE using the OpenPoly (Wang et al., 2025) dataset, another real-world benchmark. Similar to Point[2], OpenPoly contains missing property values, which were directly treated as masked inputs by our framework.

### F.1 OPENPOLY

#### F.1.1 PROPERTY PREDICTION

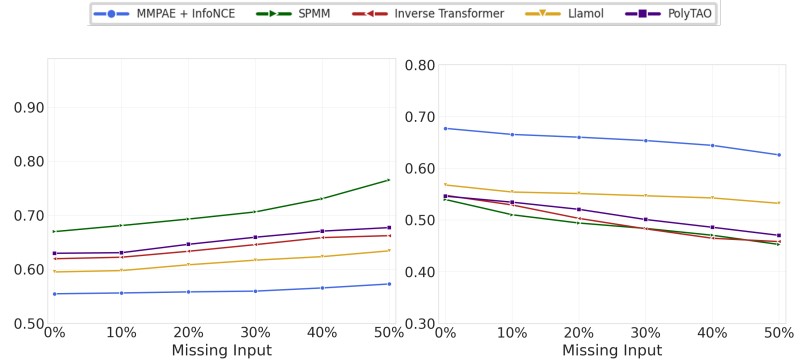

Figure 15: Property prediction result on OpenPoly dataset. The x-axis shows the fraction of missing PSMILES tokens during evaluation, and the y-axis reports RMSE ($\downarrow$).

#### F.1.2 POLYMER INVERSE DESIGN

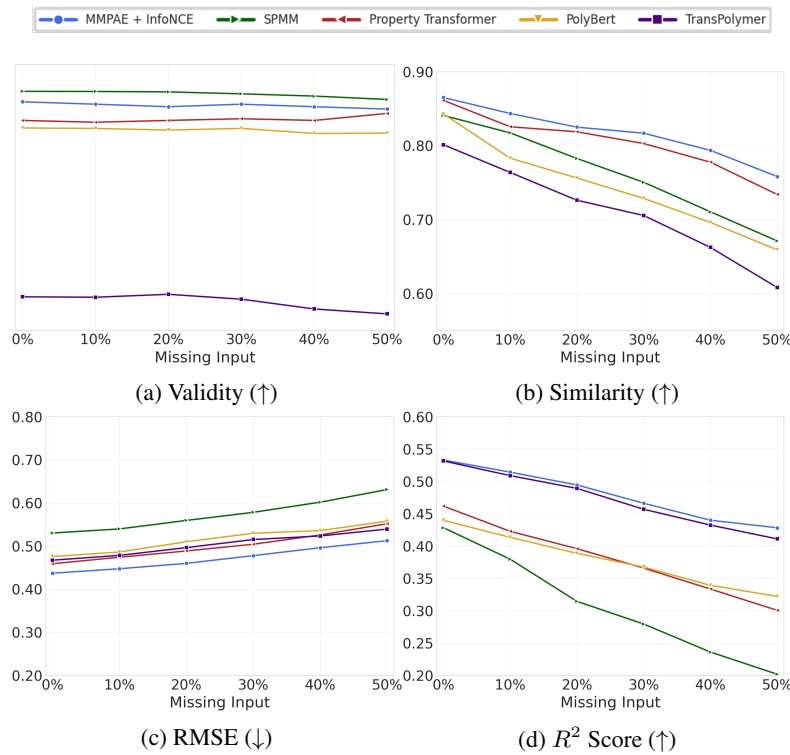

Figure 16: Inverse design result on OpenPoly dataset. The x-axis shows the fraction of missing input property values during evaluation, and the y-axis reports (a) Validity, (b) Similarity, (c) RMSE, and (d) $R2$ .

### F.1.3 CROSS-MODAL RETRIEVAL

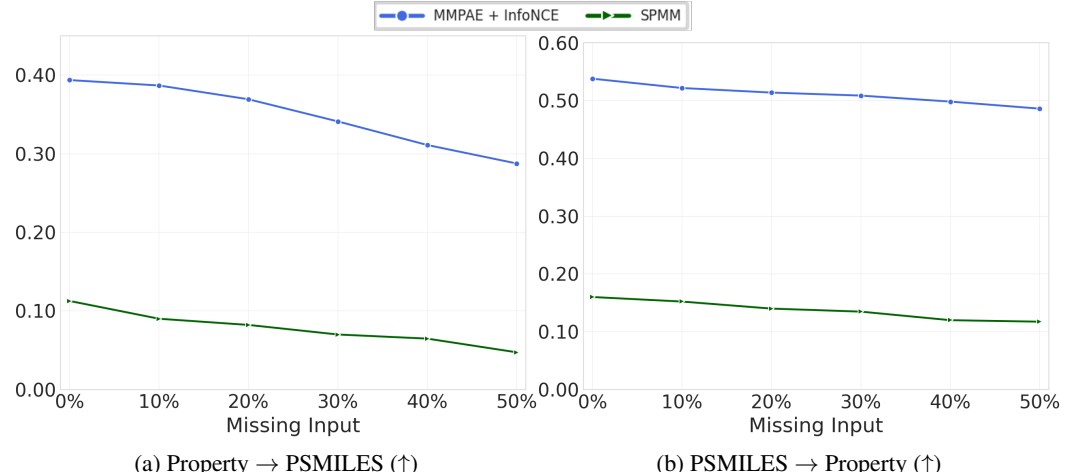

(a) Property → PSMILES (↑)

(b) PSMILES → Property (↑)

Figure 17: Top-1 accuracy of cross-modal retrieval result on OpenPoly dataset.

## G  FURTHER ABLATION STUDY

This section provides additional ablation experiments to assess the sensitivity of MMPAE to key hyperparameters. To this end, we conduct further ablations on the hyperparameters that the noticeable influence on the behavior of the our framework. In particular, we examine the InfoNCE coefficient ($\beta$) in Equation equation 9 and the temperature parameter ($\tau$) used in the contrastive objective. These analyses clarify how the components of the objective function interact under different hyperparameter settings and illustrate their respective impacts on downstream performance.

### G.1  COEFFICIENT ($\beta$) OF INFONCE

The coefficient determines the relative contribution of the InfoNCE term and thus modulates the strength of cross-modal alignment. When $\beta$ is small, the model assigns limited weight to aligning unimodal representations with the complete-submodality representation, which results in weaker performance in cross-modal retrieval and related tasks. Increasing $\beta$ strengthens alignment and leads to consistent improvements across these settings. When $\beta$ becomes excessively large, however, the alignment objective begins to compete with the reconstruction objective. This trade-off is particularly apparent in property prediction, where overly strong contrastive alignment can reduce unimodal informativeness and slightly degrade performance under fully observed inputs. Overall, $\beta$ = 10000 yields a stable balance, enhancing robustness without compromising reconstruction quality.

### G.1.1  PROPERTY PREDICTION

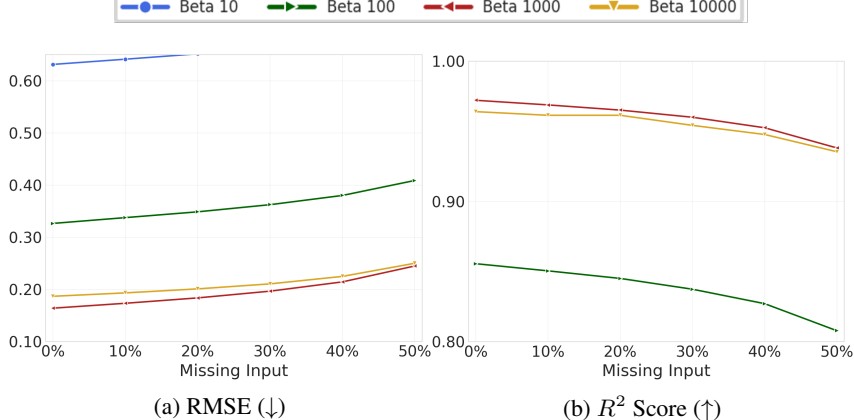

(a) RMSE ($\downarrow$)    (b) $R^2$ Score ($\uparrow$)

Figure 18: Property prediction result on complete submodality scenario. The x-axis shows the fraction of missing PSMILES tokens during evaluation, and the y-axis reports RMSE ($\downarrow$).

### G.1.2 POLYMER INVERSE DESIGN

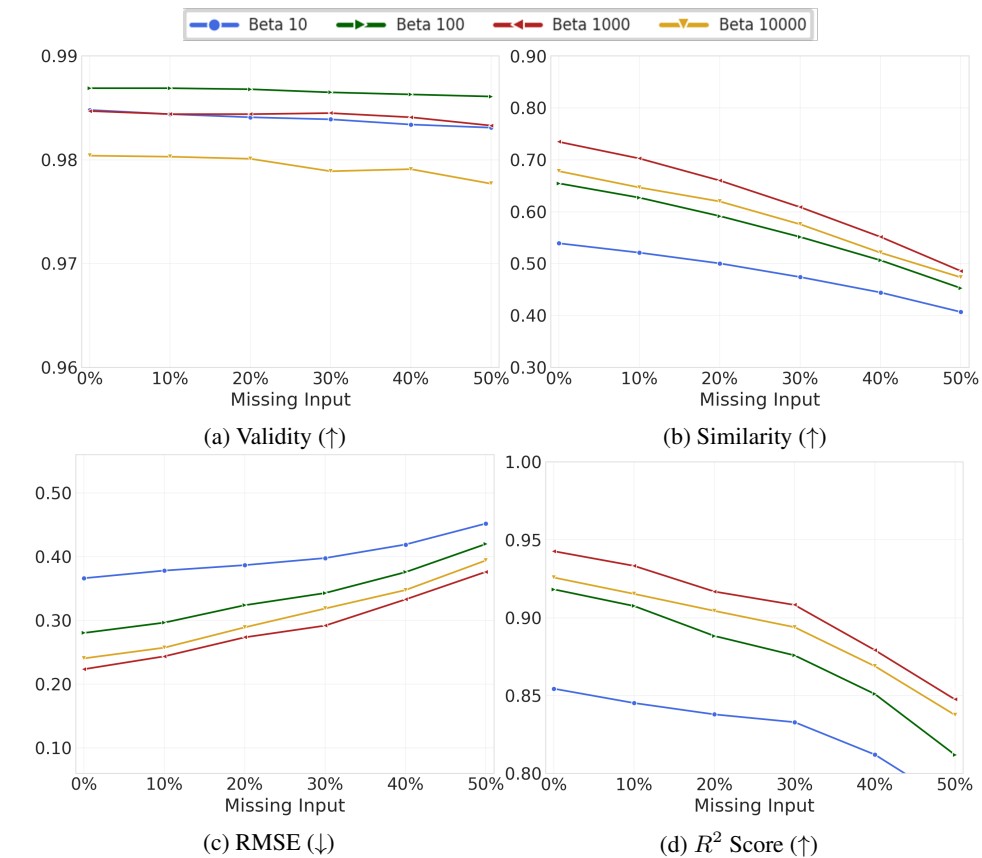

(a) Validity (↑)

(b) Similarity (↑)

(c) RMSE (↓)

(d) $R^2$ Score (↑)

Figure 19: Results in the inverse design task. The x-axis shows the fraction of missing input property values during evaluation, and the y-axis reports (a) Validity, (b) Similarity, and (c) RMSE.

### G.1.3 CROSS-MODAL RETRIEVAL

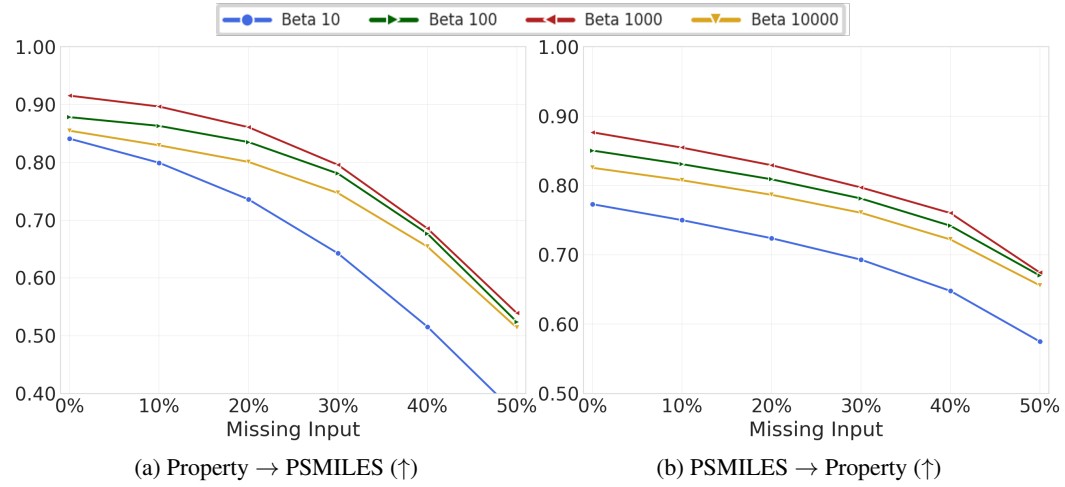

(a) Property → PSMILES (↑)

(b) PSMILES → Property (↑)

Figure 20: Results of the cross-modal retrieval task under different coefficient settings (top 1

## G.2 TEMPERATURE $\tau$ OF INFONCE

The temperature shapes the sharpness of the contrastive similarity distribution. A smaller $\tau$ increases the concentration around positive pairs, strengthening alignment but risking over-confident representations that may diminish unimodal expressiveness. Larger $\tau$ produces smoother similarity distributions but reduces the discriminative power needed for cross-modal retrieval. The empirical trends show that moderate temperatures yield the most reliable behavior across tasks. In particular, $\tau = 0.2$ consistently provides strong alignment without causing representation collapse or instability, supporting both unimodal informativeness and cross-modal consistency.

### G.2.1 PROPERTY PREDICTION

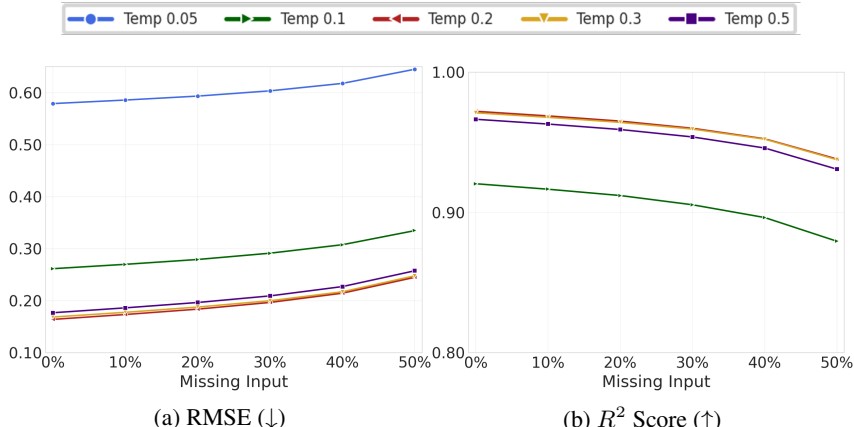

(a) RMSE ($\downarrow$)      (b) $R^2$ Score ($\uparrow$)

Figure 21: Property prediction result on complete submodality scenario. The x-axis shows the fraction of missing PSMILES tokens during evaluation, and the y-axis reports RMSE ($\downarrow$).

### G.2.2 POLYMER INVERSE DESIGN

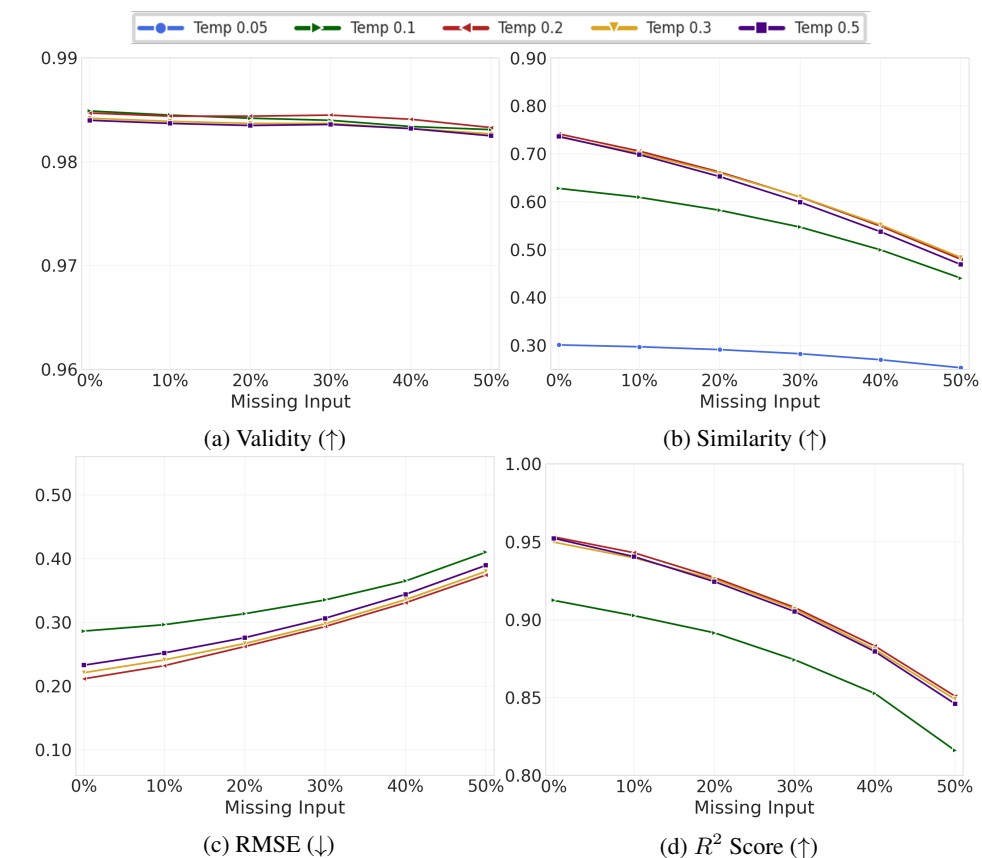

Figure 22: Results in the inverse design task. The x-axis shows the fraction of missing input property values during evaluation, and the y-axis reports (a) Validity, (b) Similarity, and (c) RMSE.

### G.2.3 CROSS-MODAL RETRIEVAL

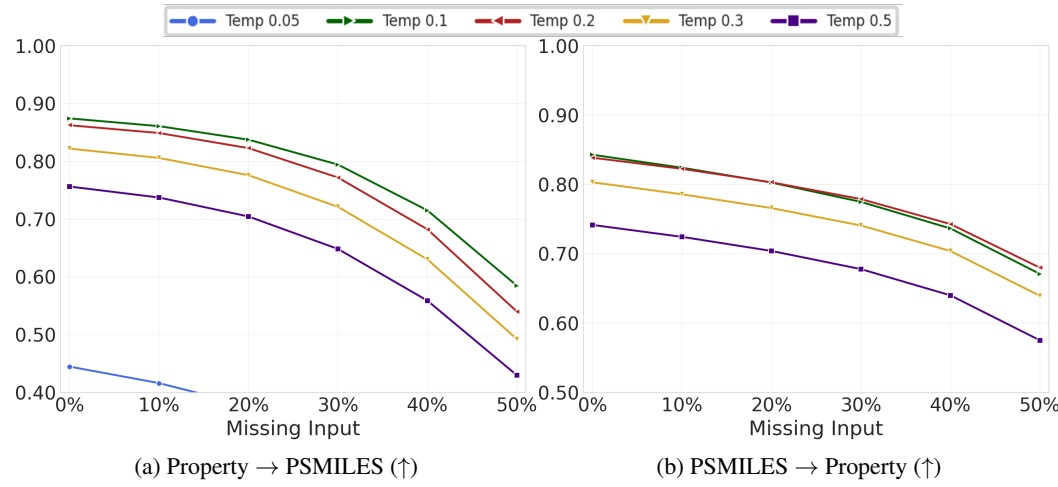

Figure 23: Results of the cross-modal retrieval task under different temperature settings (top 1

## H  QUALITATIVE EXAMPLES OF POLYMER INVERSE DESIGN

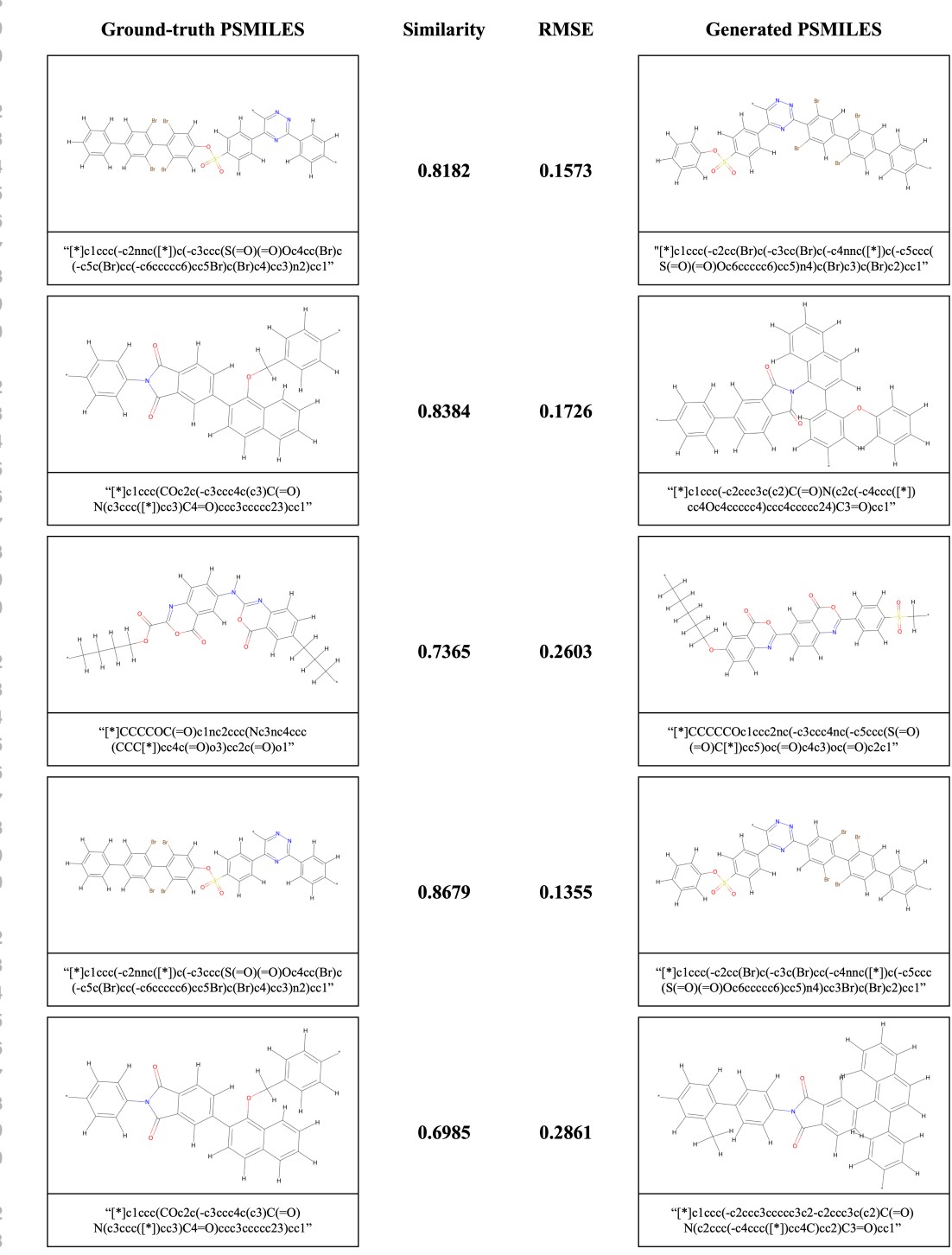

Figure 24: Qualitative examples of polymer inverse design results. Ground-truth polymers (left) and the corresponding MMPAE-generated polymers conditioned on their properties (right). Similarity and average RMSE to the target properties is shown middle in the panel and the associated PSMILES is provided below of the structures. Generated polymers exhibit structurally coherent motifs and similar RMSE trends, demonstrating that MMPAE learns meaningful structure–property relationships.

# I  PROPERTY-WISE ERROR ANALYSIS

This analysis provides property-wise RMSE, relative RMSE, and $R^2$ scores for all 29 properties. These metrics together characterize absolute accuracy, scale-normalized accuracy, and the proportion of variance in each property that is explained by the model. All metrics are computed in the original physical units of each property instead of the normalized values used during training. This ensures that reported errors correspond to physically meaningful magnitudes and remain directly interpretable within the context of polymer science. In addition, the property-wise results demonstrate that MMPAE+InfoNCE achieves the strongest overall predictive performance across all 29 properties. It obtains the lowest RMSE and lowest relative RMSE for most properties and consistently achieves very high $R^2$ values. In contrast, baseline models exhibit larger RMSE and noticeably lower $R^2$ across numerous properties. These trends indicate that MMPAE+InfoNCE provides both high accuracy and stable performance across heterogeneous physicochemical properties.

## I.1  MMPAE + INFONCE

|          | Tg      | Tm      | Td      | Cp      | Eat     | Oi      | Xc      | Xe      | rho     | Egc     | Egb     | Eea     | Ei      | Eib     | CED     |
|----------|---------|---------|---------|---------|---------|---------|---------|---------|---------|---------|---------|---------|---------|---------|---------|
| RMSE     | 5.8861  | 6.9195  | 7.2330  | 0.01063 | 0.01303 | 0.4750  | 0.9746  | 0.7693  | 0.00777 | 0.04436 | 0.04795 | 0.03906 | 0.03557 | 0.02421 | 1.9358  |
| Rel. RMSE| 0.01278 | 0.01230 | 0.01052 | 0.00815 | 0.00217 | 0.01513 | 0.02732 | 0.02272 | 0.00609 | 0.01334 | 0.01528 | 0.02000 | 0.00611 | 0.00714 | 0.01502 |
| R2       | 0.99094 | 0.98743 | 0.98480 | 0.99114 | 0.99206 | 0.99476 | 0.99281 | 0.99310 | 0.99400 | 0.99328 | 0.99306 | 0.99012 | 0.98579 | 0.99289 | 0.99024 |

|          | YM      | TSy     | TSb     | epsb    | permO2  | permCO2  | permN2   | permH2   | permHe  | permCH4  | nc      | ne      | epsc    | epse_6.0 | Avg    |
|----------|---------|---------|---------|---------|---------|----------|----------|----------|---------|----------|---------|---------|---------|----------|--------|
| RMSE     | 51.5077 | 1.6470  | 2.2119  | 2.1187  | 6.9732  | 0.35383  | 11.6717  | 3.8752   | 0.58837 | 0.00879  | 0.00501 | 0.04026 | 0.03743 | **3.3998** |
| Rel RMSE | 0.03131 | 0.02895 | 0.03261 | 0.09641 | 2.74097 | 3.20680  | 54.25507 | 0.12849  | 0.09048 | 71.68324 | 0.00452 | 0.00309 | 0.00907 | 0.01089  | **4.7323** |
| R2       | 0.97935 | 0.98049 | 0.98358 | 0.97295 | 0.99077 | 0.98909  | 0.98948  | 0.99202  | 0.99336 | 0.98300  | 0.99252 | 0.99178 | 0.99208 | 0.99034  | **0.9892** |

## I.2  SPMM

|          | Tg      | Tm      | Td      | Cp      | Eat     | Oi      | Xc      | Xe      | rho     | Egc     | Egb     | Eea     | Ei      | Eib     | CED     |
|----------|---------|---------|---------|---------|---------|---------|---------|---------|---------|---------|---------|---------|---------|---------|---------|
| RMSE     | 10.3891 | 12.1786 | 12.5373 | 0.0206  | 0.0254  | 0.9324  | 1.8949  | 1.5106  | 0.0148  | 0.0838  | 0.0908  | 0.0720  | 0.0633  | 0.0449  | 3.6323  |
| Rel RMSE | 0.0223  | 0.0215  | 0.0182  | 0.0159  | 0.0042  | 0.0299  | 0.0539  | 0.0454  | 0.0116  | 0.0256  | 0.0297  | 0.0360  | 0.0109  | 0.0133  | 0.0274  |
| R2       | 0.8736  | 0.8721  | 0.8705  | 0.8736  | 0.8740  | 0.8753  | 0.8744  | 0.8744  | 0.8749  | 0.8742  | 0.8741  | 0.8734  | 0.8709  | 0.8745  | 0.8736  |

|          | YM      | TSy     | TSb     | epsb    | permO2  | permCO2  | permN2   | permH2  | permHe  | permCH4  | nc      | ne      | epsc    | epse_6.0 | Avg    |
|----------|---------|---------|---------|---------|---------|----------|----------|---------|---------|----------|---------|---------|---------|----------|--------|
| RMSE     | 87.8156 | 2.8867  | 3.7855  | 3.4998  | 2.3512  | 13.5125  | 0.6750   | 11.6717 | 7.4634  | 1.0477   | 0.0168  | 0.0095  | 0.0776  | 0.0711   | **6.1509** |
| Rel RMSE | 0.0530  | 0.0505  | 0.0562  | 0.1838  | 2.8146  | 6.5528   | 146.1575 | 0.2799  | 0.2078  | 77.5460  | 0.0086  | 0.0059  | 0.0175  | 0.0207   | **8.0800** |
| R2       | 0.8678  | 0.8683  | 0.8695  | 0.8636  | 0.8637  | 0.8589   | 0.8640   | 0.8673  | 0.8704  | 0.8583   | 0.8742  | 0.8738  | 0.8739  | 0.8732   | **0.8707** |

## I.3  PROPERTY TRANSFORMER

|          | Tg      | Tm      | Td      | Cp      | Eat     | Oi      | Xc      | Xe      | rho     | Egc     | Egb     | Eea     | Ei      | Eib     | CED     |
|----------|---------|---------|---------|---------|---------|---------|---------|---------|---------|---------|---------|---------|---------|---------|---------|
| RMSE     | 5.9790  | 7.0624  | 7.1397  | 0.01122 | 0.01390 | 0.50467 | 1.04431 | 0.83458 | 0.00807 | 0.04792 | 0.05186 | 0.04078 | 0.03723 | 0.02521 | 2.01599 |
| Rel RMSE | 0.0223  | 0.0216  | 0.0182  | 0.0159  | 0.0042  | 0.0298  | 0.0537  | 0.0454  | 0.0115  | 0.0251  | 0.0297  | 0.0363  | 0.0109  | 0.0133  | 0.0275  |
| R2       | 0.94505 | 0.94130 | 0.93959 | 0.94453 | 0.94536 | 0.94848 | 0.94615 | 0.94628 | 0.94792 | 0.94656 | 0.94628 | 0.94363 | 0.93883 | 0.94670 | 0.94382 |

|          | YM      | TSy     | TSb     | epsb    | permO2  | permCO2  | permN2   | permH2  | permHe  | permCH4  | nc      | ne      | epsc    | epse_6.0 | Avg    |
|----------|---------|---------|---------|---------|---------|----------|----------|---------|---------|----------|---------|---------|---------|----------|--------|
| RMSE     | 52.3219 | 1.6699  | 2.2426  | 2.4237  | 1.8077  | 10.5012  | 0.54030  | 8.54046 | 5.01781 | 0.93578  | 0.00939 | 0.00533 | 0.04331 | 0.03915  | **3.8247** |
| Rel RMSE | 0.0514  | 0.0496  | 0.0549  | 0.1805  | 1.2472  | 1.4786   | 55.4742  | 0.2616  | 0.2061  | 87.3344  | 0.0087  | 0.0059  | 0.0175  | 0.0207   | **5.0606** |
| R2       | 0.93309 | 0.93434 | 0.93752 | 0.91901 | 0.93397 | 0.92967  | 0.92987  | 0.93868 | 0.94326 | 0.91141  | 0.94587 | 0.94512 | 0.94523 | 0.94383  | **0.9400** |

## I.4  POLYBERT

|          | Tg      | Tm      | Td      | Cp      | Eat     | Oi      | Xc      | Xe      | rho     | Egc     | Egb     | Eea     | Ei      | Eib     | CED     |
|----------|---------|---------|---------|---------|---------|---------|---------|---------|---------|---------|---------|---------|---------|---------|---------|
| RMSE     | 10.1272 | 11.8522 | 12.4233 | 0.0201  | 0.0246  | 0.9182  | 1.8449  | 1.4773  | 0.0144  | 0.0810  | 0.0873  | 0.0694  | 0.0611  | 0.0432  | 3.4917  |
| Rel RMSE | 0.01306 | 0.01261 | 0.01039 | 0.00863 | 0.00232 | 0.01618 | 0.02941 | 0.02477 | 0.00635 | 0.01447 | 0.01653 | 0.02089 | 0.00640 | 0.00744 | 0.01557 |
| R2       | 0.9336  | 0.9227  | 0.9159  | 0.9290  | 0.9323  | 0.9417  | 0.9351  | 0.9356  | 0.9401  | 0.9377  | 0.9365  | 0.9282  | 0.9174  | 0.9375  | 0.9284  |

|          | YM      | TSy     | TSb     | epsb    | permO2  | permCO2  | permN2   | permH2  | permHe  | permCH4  | nc      | ne      | epsc    | epse_6.0 | Avg    |
|----------|---------|---------|---------|---------|---------|----------|----------|---------|---------|----------|---------|---------|---------|----------|--------|
| RMSE     | 85.4277 | 2.8267  | 3.6800  | 3.2580  | 2.0839  | 11.1663  | 0.6117   | 10.5238 | 7.0519  | 0.8972   | 0.0163  | 0.0092  | 0.0754  | 0.0688   | **5.8701** |
| Rel RMSE | 0.03296 | 0.03044 | 0.03371 | 0.09937 | 3.69025 | 2.92726  | 76.66911 | 0.16256 | 0.12082 | 78.03120 | 0.00482 | 0.00328 | 0.00977 | 0.01140  | **5.5873** |
| R2       | 0.9022  | 0.9010  | 0.9139  | 0.8917  | 0.9094  | 0.8889   | 0.9139   | 0.9197  | 0.9316  | 0.8924   | 0.9341  | 0.9314  | 0.9322  | 0.9271   | **0.9228** |

## I.5  TRANSPOLYMER

|          | Tg      | Tm      | Td      | Cp      | Eat     | Oi      | Xc      | Xe      | rho     | Egc     | Egb     | Eea     | Ei      | Eib     | CED     |
|----------|---------|---------|---------|---------|---------|---------|---------|---------|---------|---------|---------|---------|---------|---------|---------|
| RMSE     | 10.1342 | 11.8561 | 12.3913 | 0.0202  | 0.0248  | 0.9198  | 1.8526  | 1.4883  | 0.0144  | 0.0810  | 0.0872  | 0.0694  | 0.0614  | 0.0435  | 3.4836  |
| Rel RMSE | 0.0086  | 0.0083  | 0.0073  | 0.0056  | 0.0015  | 0.0105  | 0.0189  | 0.0159  | 0.0042  | 0.0097  | 0.0112  | 0.0134  | 0.0043  | 0.0049  | 0.0102  |
| R2       | 0.9212  | 0.9114  | 0.9032  | 0.9161  | 0.9195  | 0.9282  | 0.9223  | 0.9225  | 0.9270  | 0.9253  | 0.9246  | 0.9168  | 0.9065  | 0.9253  | 0.9173  |

|          | YM      | TSy     | TSb     | epsb    | permO2  | permCO2  | permN2   | permH2  | permHe  | permCH4  | nc      | ne      | epsc    | epse_6.0 | Avg    |
|----------|---------|---------|---------|---------|---------|----------|----------|---------|---------|----------|---------|---------|---------|----------|--------|
| RMSE     | 85.6283 | 2.8204  | 3.6914  | 3.3716  | 2.0960  | 11.4708  | 0.6250   | 10.6076 | 7.0249  | 0.9716   | 0.0162  | 0.0092  | 0.0751  | 0.0687   | **5.8967** |
| Rel RMSE | 0.0210  | 0.0196  | 0.0226  | 0.0653  | 1.0698  | 2.4303   | 38.8103  | 0.2337  | 0.1430  | 137.6861 | 0.0031  | 0.0022  | 0.0063  | 0.0075   | **6.2295** |
| R2       | 0.8912  | 0.8902  | 0.9023  | 0.8804  | 0.9231  | 0.9246   | 0.9176   | 0.9248  | 0.9254  | 0.9115   | 0.9219  | 0.9196  | 0.9201  | 0.9154   | **0.9157** |

## J    DETAILED PROPERTY-WISE PREDICTION ANALYSIS

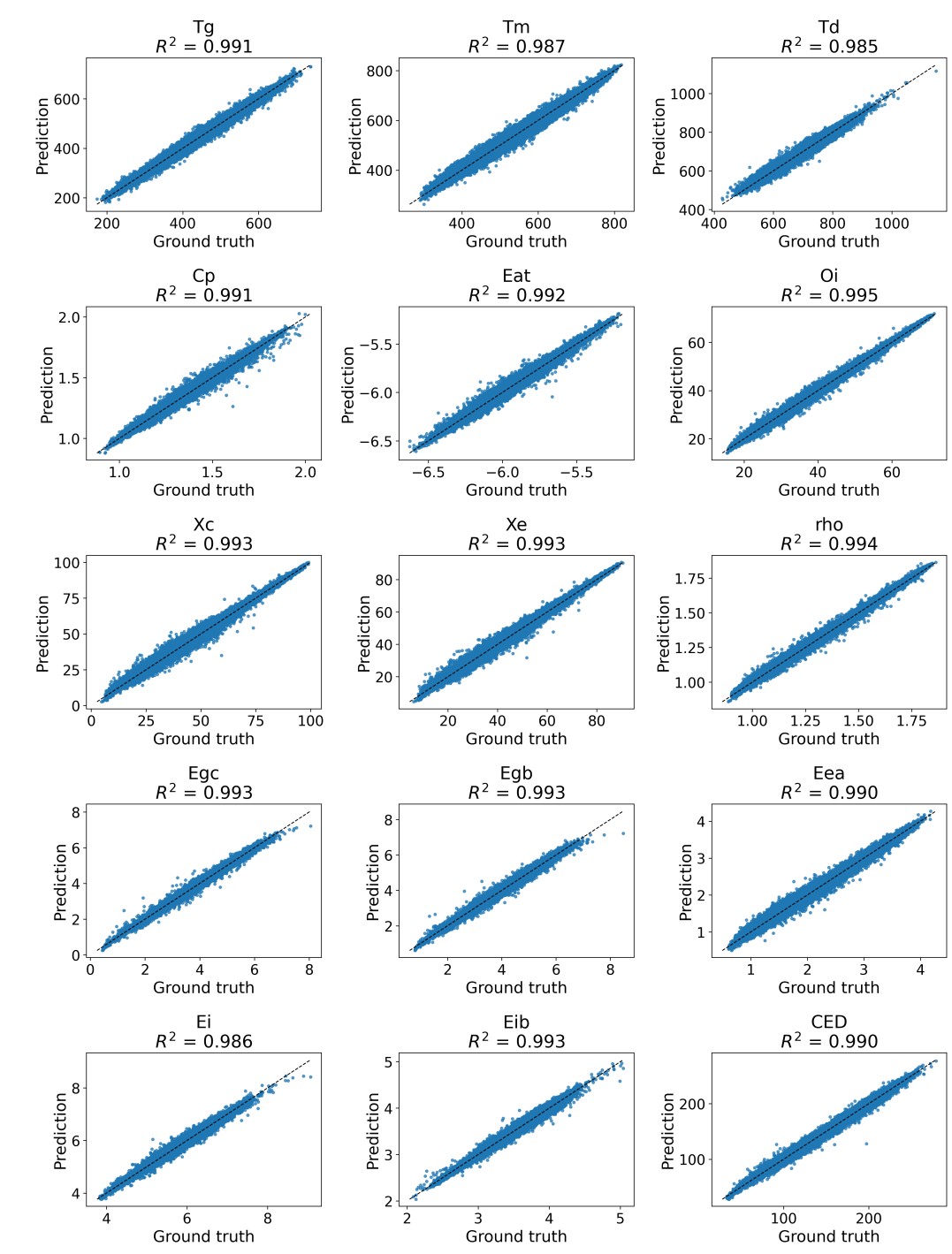

Figure 25: Scatter plots of ground-truth versus predicted values for the Thermal (Tg, Tm, Td), Thermodynamic/Physical (Cp, Eat, Oi, Xc, Xe, $\rho$), Electronic (Egc, Egb, Eea, Ei, Eib, CED) properties. Each subplot reports the property-specific coefficient of determination (R²), enabling fine-grained assessment of predictive accuracy beyond aggregate metrics. The gray dotted line is the y=x line. The close alignment with the identity line indicates that MMPAE achieves highly accurate predictions.

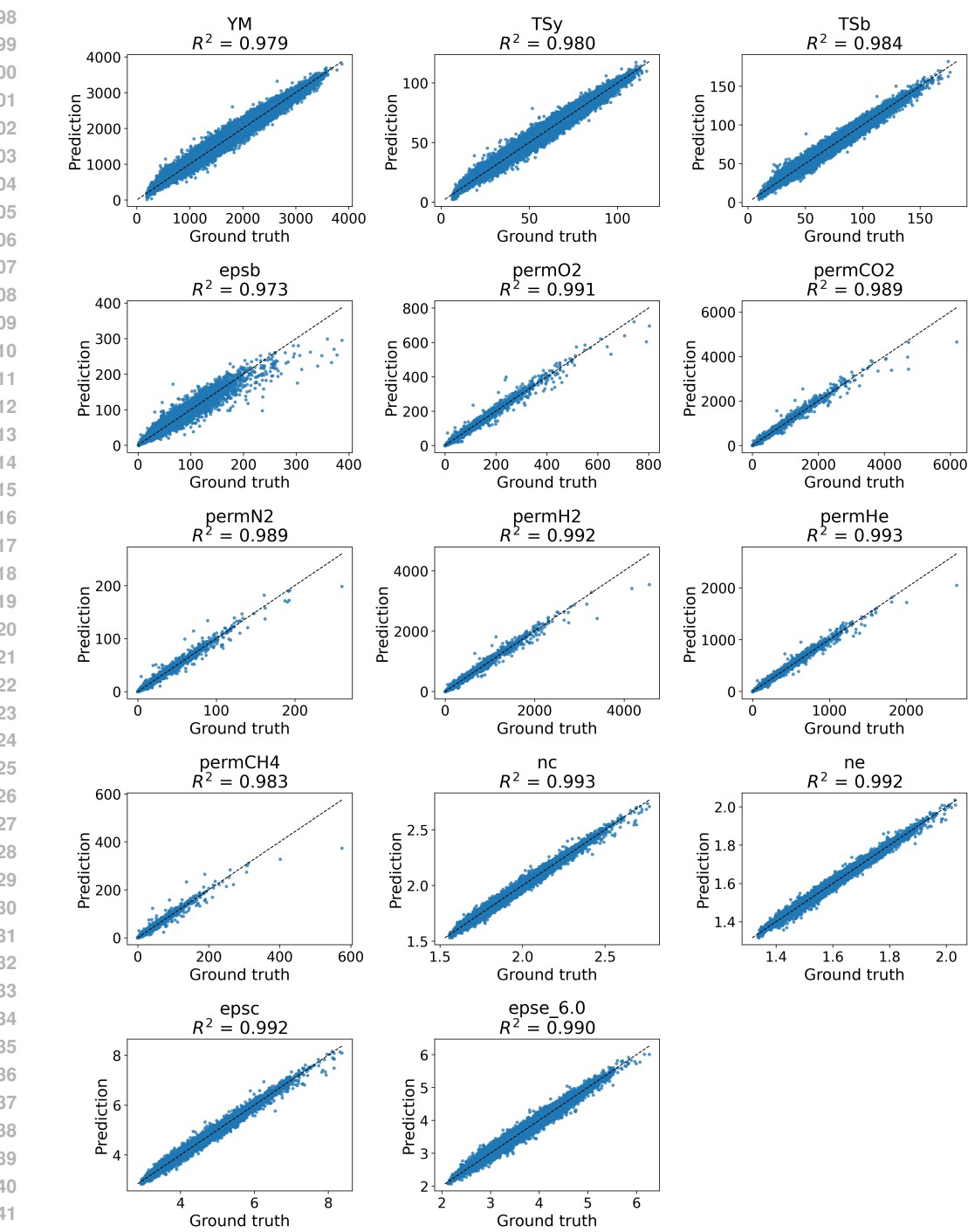

Figure 26: Scatter plots of ground-truth versus predicted values for the mechanical (YM, TSy, TSb, epsb), Permeability (permO2, permCO2, permN2, permH2, permHe, permCH4), and Optical/Dielectric (nc, ne, epsc, epse6.0) properties. Consistent with Figure 25, the results demonstrate strong property-wise prediction performance.

# K ALGORITHMIC COMPARISON OF VARIANTS OF MMPAE

## K.1 MMPAE

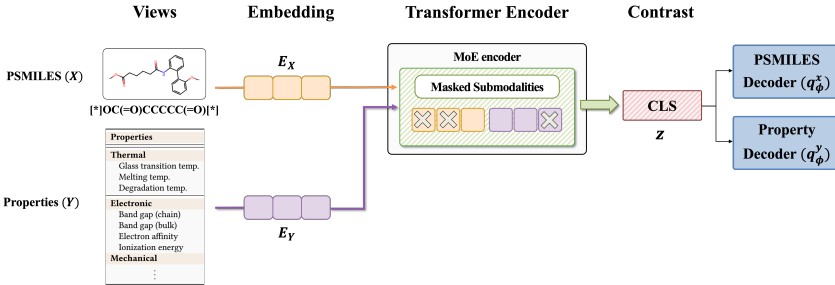

Figure 27: MMPAE optimizes the reconstruction objective in equation 2 with uniform weights on all submodality experts.

## K.2 MMPAE + HMoE

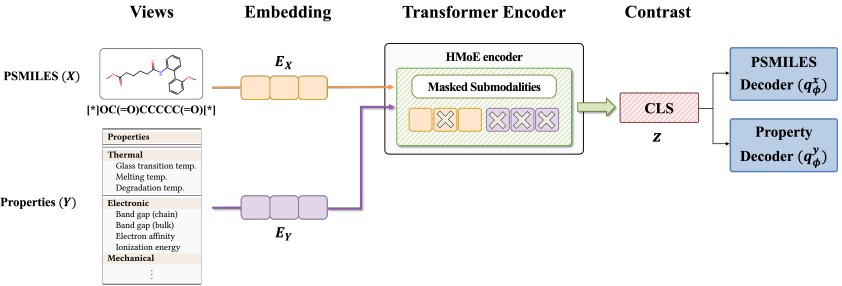

Figure 28: MMPAE+HMoE extends MMPAE by applying the hierarchical mixture-of-experts objective in Equation equation 7 to increase the weights on unimodal experts, thereby enhancing the informativeness of single-modality representations.

## K.3 MMPAE + INFONCE

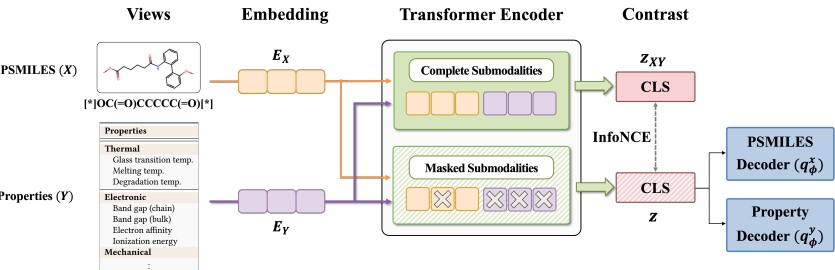

Figure 29: MMPAE+InfoNCE, our final and recommended model, further incorporates the InfoNCE objective of equation 9 to align representations across modalities while retaining the unimodal weighting of HMoE.

---

**Algorithm 1** Training process of MMPAE.

---

1: **Input:** $D$: dataset $\{x^{(i)}, y^{(i)}\}_{i=1}^{|D|}$ where $x^{(i)} = [x_1^{(i)}, \ldots, x_{|X|}^{(i)}]$, $y^{(i)} = [y_1^{(i)}, \ldots, y_{|Y|}^{(i)}]$,
   $\theta$: encoder,   $Emb^x$: PSMILES embedder,   $Emb^y$: property embedder,
   $\phi^x$: PSMILES decoder,   $\phi^y$: property deocder,   $K$: batch size,
   CE: function that returns cross-entropy loss,
   SSE: function that returns sum of square error,
   mask: function that masks each submodality in its input modality with prob. 0.5 and
       returns sequence of unmasked submodalities.
2: **for** sampled minibatch $\{x^{(k)}, y^{(k)}\}_{k=1}^{K} \sim D$ **do**
3:   **for** $k = 1$ **to** $K$ **do**
4:     ## Encode complete-submodality representation ($z_{xy}$) ##
5:     $x_{\mathrm{mask}}^{(k)} = \mathrm{mask}(x^{(k)})$,   $y_{\mathrm{mask}}^{(k)} = \mathrm{mask}(y^{(k)})$
6:     $e_{mask}^{x(k)} = Emb^x(x_{mask}^{(k)})$,   $e_{mask}^{y(k)} = Emb^y(y_{mask}^{(k)})$
7:     $z^{(k)} = \theta([e_{mask}^{x(k)}; e_{mask}^{y(k)}])$

8:     ## Decode each modality from $z$ ##
9:     $\hat{x}^{(k)} = \phi^x(z)$,   $\hat{y}^{(k)} = \phi^y(z)$.
10:   **end for**
11:   **define** $f(a, b) := \langle a, b \rangle$
12:   **for** $k = 1$ **to** $K$ **do**
13:     $L_{recon}^{(k)} = \mathrm{CE}(x^{(k)}, \hat{x}^{(k)}) + \mathrm{SSE}(y^{(k)}, \hat{y}^{(k)})$
14:   **end for**
15:   Update $\theta, Emb^x, Emb^y, \phi^x, \phi^y$ to minimize $L = \frac{1}{K} \sum_{k=1}^{K} \left[ L_{recon}^{(k)} \right]$
16: **end for**

---

---

**Algorithm 2** Training process of MMPAE + HMoE.

---

1: **Input:** $D$: dataset $\{x^{(i)}, y^{(i)}\}_{i=1}^{|D|}$ where $x^{(i)} = [x_1^{(i)}, \ldots, x_{|X|}^{(i)}]$, $y^{(i)} = [y_1^{(i)}, \ldots, y_{|Y|}^{(i)}]$,
$\quad\quad\quad\theta$: encoder, $\quad Emb^x$: PSMILES embedder, $\quad Emb^y$: property embedder,
$\quad\quad\quad\phi^x$: PSMILES decoder, $\quad \phi^y$: property deocder, $\quad K$: batch size,
$\quad\quad\quad$CE: function that returns cross-entropy loss,
$\quad\quad\quad$SSE: function that returns sum of square error,
$\quad\quad\quad$mask: function that masks each submodality in its input modality with prob. 0.5 and
$\quad\quad\quad\quad\quad$returns sequence of unmasked submodalities.
2: **for** sampled minibatch $\{x^{(k)}, y^{(k)}\}_{k=1}^K \sim D$ **do**
3: $\quad$ **for** $k = 1$ **to** $K$ **do**
4: $\quad\quad$ ## Encode modality-specific representation ($z$) via HMoE ##
5: $\quad\quad$ $x_{\text{mask}}^{(k)} = \text{mask}(x^{(k)}), \quad y_{\text{mask}}^{(k)} = \text{mask}(y^{(k)})$
6: $\quad\quad$ $e_{mask}^{x(k)} = Emb^x(x_{mask}^{(k)}), \quad e_{mask}^{y(k)} = Emb^y(y_{mask}^{(k)})$
7: $\quad\quad$ sample $m^{(k)} \sim \text{Bernoulli}(0.5)$
8: $\quad\quad$ $z^{(k)} = \theta^{\text{HMoE}}(e_{mask}^{x(k)}, e_{mask}^{y(k)}) = \begin{cases} \theta(e_{mask}^{x(k)}) & \text{if } m^{(k)} = 1 \\ \theta(e_{mask}^{y(k)}) & \text{otherwise} \end{cases}$

9: $\quad\quad$ ## Decode each modality from $z$ ##
10: $\quad\quad$ $\hat{x}^{(k)} = \phi^x(z), \quad \hat{y}^{(k)} = \phi^y(z)$.
11: $\quad$ **end for**
12: $\quad$ **define** $f(a, b) := \langle a, b \rangle$
13: $\quad$ **for** $k = 1$ **to** $K$ **do**
14: $\quad\quad$ $L_{recon}^{(k)} = \text{CE}(x^{(k)}, \hat{x}^{(k)}) + \text{SSE}(y^{(k)}, \hat{y}^{(k)})$
15: $\quad$ **end for**
16: $\quad$ Update $\theta, Emb^x, Emb^y, \phi^x, \phi^y$ to minimize $L = \frac{1}{K} \sum_{k=1}^K \left[ L_{recon}^{(k)} \right]$
17: **end for**

---

---

**Algorithm 3** Training process of MMPAE + InfoNCE.

---

1: **Input:** $D$: dataset $\{x^{(i)}, y^{(i)}\}_{i=1}^{|D|}$ where $x^{(i)} = [x_1^{(i)}, \ldots, x_{|X|}^{(i)}]$, $y^{(i)} = [y_1^{(i)}, \ldots, y_{|Y|}^{(i)}]$,

   $\theta$: encoder, $Emb^x$: PSMILES embedder, $Emb^y$: property embedder,

   $\phi^x$: PSMILES decoder, $\phi^y$: property deocder, $K$: batch size,

   CE: function that returns cross-entropy loss,

   SSE: function that returns sum of square error,

   mask: function that masks each submodality in its input modality with prob. 0.5 and

     returns sequence of unmasked submodalities.

2: **for** sampled minibatch $\{x^{(k)}, y^{(k)}\}_{k=1}^K \sim D$ **do**

3: **for** $k = 1$ **to** $K$ **do**

4:  ## Encode complete-submodality representation ($z_{xy}$) ##

5:  $e^{x(k)} = Emb^x(x^{(k)})$, $e^{y(k)} = Emb^y(y^{(k)})$

6:  $z_{xy}^{(k)} = \theta([e^{x(k)}; e^{y(k)}])$

7:  ## Encode modality-specific representation ($z$) via HMoE ##

8:  $x_{\text{mask}}^{(k)} = \text{mask}(x^{(k)})$, $y_{\text{mask}}^{(k)} = \text{mask}(y^{(k)})$

9:  $e_{mask}^{x(k)} = Emb^x(x_{mask}^{(k)})$, $e_{mask}^{y(k)} = Emb^y(y_{mask}^{(k)})$

10:  sample $m^{(k)} \sim \text{Bernoulli}(0.5)$

11:  $z^{(k)} = \theta^{\text{HMoE}}(e_{mask}^{x(k)}, e_{mask}^{y(k)}) = \begin{cases} \theta(e_{mask}^{x(k)}) & \text{if } m^{(k)} = 1 \\ \theta(e_{mask}^{y(k)}) & \text{otherwise} \end{cases}$

12:  ## Decode each modality from $z$ ##

13:  $\hat{x}^{(k)} = \phi^x(z)$, $\hat{y}^{(k)} = \phi^y(z)$.

14: **end for**

15: **define** $f(a, b) := \langle a, b \rangle$

16: **for** $k = 1$ **to** $K$ **do**

17:  $L_{recon}^{(k)} = \text{CE}(x^{(k)}, \hat{x}^{(k)}) + \text{SSE}(y^{(k)}, \hat{y}^{(k)})$

18:  $L_{InfoNCE}^{(k)} = -\log \dfrac{e^{f\left(z^{(k)}, z_{xy}^{(k)}\right)}}{\sum_{j=1}^K e^{f\left(z^{(k)}, z_{xy}^{(j)}\right)}}$

19: **end for**

20: Update $\theta, Emb^x, Emb^y, \phi^x, \phi^y$ to minimize $L = \frac{1}{K} \sum_{k=1}^K \left[ L_{recon}^{(k)} + \beta \cdot L_{InfoNCE}^{(k)} \right]$

21: **end for**

---

## L    THE USE OF LARGE LANGUAGE MODELS (LLMS)

We utilized ChatGPT as an assistive tool in this work. Its use was limited to (1) revising drafts and correcting grammar, and (2) conducting preliminary searches for related literature. Notably, we emphasize that all core ideas and contributions presented herein are solely the work of the authors.

