# OpenReview forum: "Multimodal Masked Polymer Autoencoder for Unified Polymer Informatics"
_ICLR.cc/2026/Conference — Submitted to ICLR 2026_

### Official Review · Reviewer_QhSn · 2025-10-21

**Soundness:** 2
**Presentation:** 1
**Contribution:** 3
**Rating:** 2
**Confidence:** 4

**Summary:**

This paper proposes a multimodal representation framework that treats polymer structures and numerical properties within the shared latent space.
In particular, each attribute is treated as an individual submodality, while hierarchical mixture-of-experts reweighting and the InfoNCE alignment are further incorporated to improve performance.
Extensive experiments demonstrate that the proposed method achieves strong performance across various tasks.

**Strengths:**

1. Through bridging structure and property within a single end-to-end model, MMPAE provides a flexible foundation for polymer informatics.
2. To overcome the limitations of the straightforward masked reconstruction objective, this paper further incorporates adaptive unimodal weighting and explicit cross-modal alignment, thereby yielding more balanced representations and improving cross-modal task performance.
3. Extensive experiments on various tasks demonstrate the effectiveness of the proposed method.

**Weaknesses:**

1. The claimed “multimodal” setup is very unconvincing.
    * MMPAE only uses polymer structures and numerical properties as input, closer to multi‑view learning than multimodality.
    * The multimodal methods mentioned in the paper integrate complementary sources of information, such as 2D graphs and 3D geometries, which differ fundamentally from the setup in this work. Therefore, referencing these studies to justify MMPAE is inappropriate.

2. The experimental section is very problematic.
    * For the dataset used in this work, polymers within this dataset are generated by enumeratively combining chemical fragments extracted from synthesized polymers, and their properties are predicted by PolyBERT [1] rather than experimental measurements or high‑fidelity simulations. Such a dataset may be suitable for pretraining purposes (as in PolyBERT itself), but should not be used for benchmarking or evaluating model performance, as it cannot provide reliable or meaningful validation of the proposed method.
    * For the baselines used in this work, it's necessary to include more recent and competitive methods, such as MMPolymer [2], Uni-Poly [3], PolyNC [4], and MCP [5]. The current baselines (e.g., Transpolymer and PolyBERT) are outdated, making the comparisons unconvincing and the claimed performance improvements questionable.
    * For the settings used in this work, randomly masking the input is quite questionable and scientifically unsound. For example, when PSMILES tokens are randomly masked, the ground‑truth property of the original polymer may no longer hold, as the modified polymer no longer corresponds to the same chemical structure. In this case, the experimental results lose their validity and persuasiveness.
    * In addition, it is necessary to provide the original numerical experimental results rather than only presenting visualized figures, as the latter cannot fully support quantitative.

[1] Kuenneth C, Ramprasad R. polyBERT: a chemical language model to enable fully machine-driven ultrafast polymer informatics[J]. Nature communications, 2023, 14(1): 4099.

[2] Wang F, Guo W, Cheng M, et al. Mmpolymer: A multimodal multitask pretraining framework for polymer property prediction[C]//Proceedings of the 33rd ACM International Conference on Information and Knowledge Management. 2024: 2336-2346.

[3] Huang Q, Li Y, Zhu L, et al. Unified multimodal multidomain polymer representation for property prediction[J]. npj Computational Materials, 2025, 11(1): 153.

[4] Qiu H, Sun Z Y. On-demand reverse design of polymers with PolyTAO[J]. npj Computational Materials, 2024, 10(1): 273.

[5] Zhang, Yipeng, Cong Shen, and Kelin Xia. "Multi-Cover Persistence (MCP)-based machine learning for polymer property prediction." Briefings in Bioinformatics 25.6 (2024): bbae465.

**Questions:**

Could you provide specific algorithms illustrating the training and inference process of MMPAE, MMPAE+HMoE, and MMPAE+InfoNCE?
Providing these algorithms would help better illustrate the distinctions among the three variants.

---

> ### Author Response · Authors · 2025-11-21
> **Rebuttal by Authors**
>
> **W1. Issues in multimodal setup**
>
> Multimodal is generally defined by the integration of heterogeneous information sources that differ in data type, statistical characteristics, and semantic meaning. In contrast, multi-view learning typically refers to multiple views of the same modality (e.g., front and side image of the same object in computer vision).
>
> In our setting, PSMILES and polymer properties satisfy the criterion of heterogenity: the former is a symbolic sequence encoding structural topology, whereas the latter is a numerical vector describing macroscopic physicochemical responses.  These two source of differ in representation form, physical origin, and semantic role, and this distinction aligns with established multimodal formulations used in polymer informatics, where structure–property pairs are treated as distinct modalities [1–3].
> In addition, the multimodal approaches cited in the manuscript were referenced not to claim identical modality configuration or identical objectives, but to motivate the need for multimodal learning in polymer domain.
>
> [1] Bidirectional generation of structure and properties through a single molecular foundation model (Chang & Ye, 2024)
>
> [2] Unified multimodal multidomain polymer representation for property prediction (Qi Huang, 2025)
>
> [3] PolyNC: a natural and chemical language model for the prediction of unified polymer properties (Haoke Qiu et al., 2023)
>
>
> **W2-1. Limitations of the PolyOne dataset**
>
> First, we would like to clarify the broader context of polymer informatics, in which experimentally measured polymer datasets are difficult and expensive to obtain at scale, and publicly available realistic datasets remain limited [1]. Consequently, many studies in polymer informatics rely on simulation- or DFT-based datasets, and it is common to use model-generated or rule-based hypothetical datasets to enable large-scale training and benchmarking [2–4].
>
> To address the concern regarding real-world applicability, we additionally train and evaluate MMPAE and all baseline models on two realistic datasets, Point2 [5] and OpenPoly [6], both of which provide real polymer structures paired with experimentally measured properties.
> As is typical for experimentally curated polymer datasets, these datasets contain missing property values; we treat such missing entries as masked submodalities under our multimodal framework.
>
> The full evaluation under varying missing input conditions is provided in Appendix E.
>
> Despite the limited size and the presence of substantial missing values, MMPAE demonstrates strong and consistent performance across nearly all metrics. This indicates that the model effectively exploits multimodal interactions between PSMILES and properties, capturing complementary modality information.
>
> Overall, MMPAE exhibits strong performance on real polymer data and consistently outperforms baseline models across all tasks, addressing the concern about its practical applicability.
>
> [1] Machine Learning in Polymer Research (W. Ge., et al, 2025)
>
> [2] Generation of a large‑scale polymer library using rule‑based generative methods (Yue et al., 2024)
>
> [3] High‑Throughput Screening and Prediction of Polymers (T. Yue et al., 2023)
>
> [4] A graph representation of molecular ensembles for polymer property prediction (Aldeghi & Coley, 2022)
>
> [5] Point^2: A Polymer Informatics Training and Testing Database (J. Xu et al., 2025)
>
> [6] OpenPoly: A Polymer Database Empowering Benchmarking and Multi-property Predictions (J Wang et al., 2025)

---

> ### Author Response · Authors · 2025-11-21
> **Rebuttal by Authors**
>
> **W2-2. Baseline selection and competitiveness**
>
>
> The suggested methods (MMPolymer, Uni-Poly, PolyNC, MCP) cannot be used as baselines in our setting because their required input modalities are fundamentally incompatible with the experimental protocol of MMPAE.
>
> Our evaluation assumes access only to 1D PSMILES and numerical properties. Comparing our model against methods that require richer or additional modalities would not constitute a fair or meaningful comparison.
>
> Below, we summarize why each method is incompatible with our setup:
>
> **MMPolymer** requires 3D conformation as a core modality; its all pretraining, representation learning, and downstream task all rely on 3D geometric embeddings in conjunction with PSMILES. Since our benchmark provides only 1D PSMILES and numerical properties, MMPolymer cannot serve as baseline.
>
> **Uni-Poly** is a five-modality framework that integrates SMILES, 2D graphs, 3D geometries, textual descriptions, and fingerprints through modality-specific encoders and a late-fusion architecture. It cannot be trained or evaluated when only PSMILES is available.
>
> **PolyNC** is a text-to-text framework that requires natural-language property descriptions in addition to PSMILES. Therefore, it is not applicable to direct numerical property regression from PSMILES alone.
>
> **MCP** depends on 3D conformations to construct topological descriptors (e.g., Delaunay-based persistence features) as essential inputs. Without 3D coordinates, MCP cannot be executed under our input regime.
>
> In summary, these methods rely on fundamentally different modality configurations or task assumptions and therefore cannot be meaningfully compared within the modality constraints defined in our study.
>
>
> **W2-3. Validity of the random masking protocol**
>
> Our token-level masking evaluation is intended to measure the sensitivity of  a property prediction model to partially observable structural information. Masking individual PSMILES tokens offers a controlled mechanism for reducing structural cues, allowing us to assess whether the model can still infer relevant property signals under incomplete inputs. This setup aligns with the masked-token modeling protocols widely used in polymer sequence modeling [1-3].
>
> We note, however, that in practical scenarios, missing structural information often occurs at the chemical-fragment level rather than at individual tokens. To reflect a more practical and structured form of incompleteness, we additionally conduct a fragment-level masking evaluation using BRICS decomposition[4], where an entire chemically meaningful fragment is removed at random.
>
>
> | **Model**               | **RMSE (↓)** | **R² (↑)**   |
> |---------------------|----------|----------|
> | **MMPAE + InfoNCE**     | **0.1357**   | **0.9805**   |
> | Property Transformer | 0.1602  | 0.9661   |
> | SPMM                | 0.2974   | 0.9018   |
> | PolyBert            | 0.1987   | 0.9584   |
> | TransPolymer        | 0.2034   | 0.9537   |
>
>
> These results show that MMPAE maintains high predictive accuracy even under fragment-level masking, demonstrating that the model remains robust when chemically meaningful substructures are removed.
>
> We note that the other baselines also exhibit reasonable robustness under fragment-level masking. This is likely because the masking protocol exposes all models to diverse combinations of partially observed PSMILES submodalities, enabling them to retain informative structure cues even when chemically meaningful fragments are removed.
>
> Overall, these findings confirm that the token-level masking protocol remains reliable, as its conclusions are consistent with those observed under the more realistic fragment-level missing-structure setting.
>
> 1] polyBERT: a chemical language model to enable fully machine-driven ultrafast polymer informatics (Christopher Kuenneth & Rampi Ramprasad, 2023)
>
> [2] TransPolymer: a Transformer-based language model for polymer property predictions (C. Xu et al., 2023)
>
> [3] MMPolymer: A Multimodal Multitask Pretraining Framework for Polymer Property Prediction (F. Wang et al., 2024)
>
> [4] On the art of compiling and using 'drug-like' chemical fragment spaces (D Jorg et al., 2008)
>
>
>
> **Q1. Illustrating the framework variants**
>
> To clarify the distinctions among the three variants, the revised manuscript includes variants-specific diagrams in Appendix J to provide algorithmic descriptions of all MMPAE variants together with corresponding schematic illustrations. These illustrations highlight how the baseline MMPAE performs masked reconstruction, how HMoE introduces modality-specific and joint experts, and how InfoNCE incorporates cross-modal alignment during training. We expect these additions to make the differences among the variants more explicit.

---

> > ### Comment · Reviewer_QhSn · 2025-11-27
> >
> > Thanks for your detailed responses, which have addressed most of my concerns.
> >
> > However, I maintain my reservation regarding the claimed “multimodal” nature of this paper, as classifying numerical properties as a modality appears tenuous and creates unnecessary ambiguity.
> >
> > Could you provide more related work supporting this classification?

---

> > > ### Author Response · Authors · 2025-11-28
> > > **Follow-up Response by Authors**
> > >
> > > First, the information encoded in numerical properties is fundamentally heterogeneous from structural inputs.
> > > These attributes capture physicochemical, thermodynamic, and topological signals that are not obtainable from PSMILES alone. Because they provide an independent source of information, numerical properties can be treated as a standalone modality in a multimodal formulation.
> > >
> > > Several prior works adopt this perspective:
> > > *  **Add-GNN [1]** treats numerical properties as an independent modality, processed through a dedicated channel separate from the structural graph and fused at the representation level. This explicitly treats descriptors as a distinct informational modality.
> > > * **Deep Learning–Driven Multimodal Integration [2]** encodes tabular numerical properties independently from text and image inputs and combines them only during multimodal fusion, establishing property values as a standalone modality.
> > > * **Multimodal NMR-driven Polymer Modeling [3]** jointly integrates scalar polymer properties with NMR spectra and structural descriptors, where numerical properties act as their own dedicated modality within the multimodal pipeline.
> > >
> > > Taken together, these studies demonstrate that numerical property vectors are consistently modeled as a heterogeneous information source and thus constitute a valid modality in multimodal polymer and materials informatics.
> > >
> > > We have incorporated these references and clarifications to the revised manuscript.
> > >
> > > [1] Add-GNN: A Dual-Representation Fusion Molecular Property Prediction Based on Graph Neural Networks with Additive Attention (R. Zhou et al., 2025)
> > >
> > > [2] Deep Learning-Driven Integration of Multimodal Data for Material Property Predictions (V. Costa et al., 2025)
> > >
> > > [3] Simultaneous multimodal and multitask strategies for diverse biodegradable polymers powered by NMR data science (X. Ni et al ,. 2025)

---

### Official Review · Reviewer_yNFG · 2025-10-24

**Soundness:** 2
**Presentation:** 1
**Contribution:** 2
**Rating:** 2
**Confidence:** 3

**Summary:**

This work proposes the MMPAE that can unify cross-modal generation/ retrieval of polymer structures and their properties. This model uses a hierarchical MOE to weight unimodal and joint contributions to overcome the flaws from straightforward masked reconstruction training. They mainly have two novelties. One is the hierarchical MOE encoder with a mutual information objective to encourage balanced informativeness across inputs. The other one is the framework that treats each feature as an individual submodality, then uses a transformer for cross-model generation and retrieval. Experiment results show MMPAE outperforms existing multimodal approaches, as well as the strong task-specific baselines in some tasks.

**Strengths:**

1. The theoretical proofs in the work are comprehensive. In Section 3, they have proved the limitation in optimizing with the reconstruction objective using mutual information-related methods. Also, they have inferred the final training objective in equation 9 step by step. I admire the author's mathematical knowledge.
2. This work explored the combination of PSMILES and properties and gained good performance compared with baselines.

**Weaknesses:**

1. I am throwing out a question here: the submodality defined in this paper corresponds to the polymor patch or each property. It is like tokens in the transformer models; it is a sequential data format that represents their modality altogether. In my opinion, this cannot be claimed as a multimodal model, as it does not handle two types of input for the same entity; there is no explicit fusion or alignment design for modalities.
2. For the model structure, it uses a typical auto-encoder transformer model, and uses the CLS token for the two downstream tasks. The novelty may only be slight changes to the loss design based on the MI, which are not significant enough.
3. In the Figure 2 training approach, the upper CLS is from unmasked embeddings from both X and Y, the lower CLS is from masked ones (after encoding by the transformer). The contrastive learning is conducted between these two CLS tokens, which will lead the model to learn an objective -- how to predict CLS better with less information. In this way, the more input missing, the larger the margin should appear between MMPAE and baselines; however, we did not see this pattern in the results, Figures 3 and 4.
4. There are no ablation experiments in this paper, which can not prove the methods and novelties proposed in Section 3.
5. The writing of this paper can be improved. The font in the figure is not consistent with the main text. The motivation claim in the introduction is not well organized and clear, which takes a longer time for readers to follow.

**Questions:**

1. Is it a common way to define one property as one modality in the polymer field? What's the rationale for defining in this way?
2. The method in Figure 2 shows using only the CLS to reconstruct the PSMILES, instead of using a list of tokens in Figure 1. How to decode the whole PSMILES with only one CLS token?

---

> ### Author Response · Authors · 2025-11-21
> **Rebuttal by Authors**
>
> **W1. Validity of the multimodal formulation**
>
> Firstly, we clarify that our framework indeed operates on two distinct modalities: the PSMILES string and the associated property vector. The PSMILES sequence encodes monomer-level structural information through discrete tokens, whereas the property vector captures macroscopic physicochemical attributes as continuous numerical features. Treating these as heterogeneous modalities is consistent with prior multimodal molecular studies, such as SPMM [1], which explicitly treat SMILES strings and property vectors as separate modalities within a unified representation framework.
>
> In addition, our method conducts explicit multimodal fusion and alignment. The architecture adopts an early-fusion scheme in which modality-specific embeddings are concatenated and jointly processed by a single Transformer encoder, similar to the design used in prior multimodal masked autoencoder frameworks [2–3]. Beyond fusion, MMPAE incorporates explicit cross-modal alignment: the HMoE encoder produces modality-specific representations for structure and property, and these are aligned with the complete submodality representation via the InfoNCE objective. This encourages both modalities to interact coherently within a shared latent space.
>
> Regarding the definition of submodalities, each modality naturally decomposes into heterogeneous and semantically meaningful units. PSMILES tokens correspond to heterogeneous structural elements (e.g., atoms, branching symbols, ring indicators) [4], and individual property reflects a heterogeneous physicochemical attribute of the polymer (thermal, mechanical, electronic, etc.) [5]. Since neither modality is homogeneous, treating these units as submodalities is appropriate and essential for modeling realistic missing-input scenarios, where some structural cues or property dimensions may be absent.
>
> [1] Bidirectional generation of structure and properties through a single molecular foundation model (Chang & Ye, 2024)
>
> [2] Multimodal Masked Autoencoders Learn Transferable Representations (X Geng et al., 2022)
>
> [3] Multimodal Channel-Mixing: Channel and Spatial Masked AutoEncoder on Facial Action Unit Detection (X. Zhang et al., 2022)
>
> [4] Hybrid fragment-SMILES tokenization for ADMET prediction in drug discovery (N. Aksamit et al., 2024)
>
> [5] Estimation and Prediction of the Polymers’ Physical Characteristics Using the Machine Learning Models (I. V. Malashin et al., 2023)
>
>
> **W2. Novelty of the proposed method**
>
> We strongly believe that our framework provides a principled approach to identifying and addressing a fundamental limitation of naive multimodal masking models. Specifically, our objective function is not a simple modification of existing approaches but a non-trivial formulation grounded in an information-theoretic perspective.
>
> When the individual structure tokens and properties are independently masked and reconstructed, the model tends to learn imbalanced informativeness between the modal-specific and the cross-modal representations (see Sec 3.2). This imbalance degrades unimodal downstream performance and is not resolved by standard reconstruction objectives or conventional MI-based regularization.
>
> To address this issue, we introduce a hierarchical mixture-of-experts (HMoE) that explicitly controls the weights of unimodal and multimodal experts, thereby enhancing the informativeness of the modal-specific representations. Furthermore, as detailed in Sec 3.3, combining the HMoE reconstruction objective with the InfoNCE alignment yields consistent empirical improvements across all downstream tasks.
>
> These contributions constitute the core novelty of our method: we provide both a theoretical identification of an overlooked limitation in multimodal masking models and a principled objective formulation to resolve it, rather than changes in architecture components.

---

> > ### Comment · Reviewer_yNFG · 2025-11-26
> > **Reply to Rebuttal Weakness 1 and 2**
> >
> > - Firstly, I agree with the author that they have combined two main modalities. After seeing the explanation, I still disagree with the paper about the sub-modality definition. I think a token or property can not be explained as a modality rather than a feature.
> > - Secondly, the author highlights the HMoe design for the main novelty. I read Section 3.3 again. If I understand correctly, the $x$ and $y$ mean two modal inputs. In line 284, the paper set $\lambda^{xy}$ to 0. Then, what's the meaning of this term?
> > - Thirdly, as an important novelty, I fail to see the module in both Figure 1 and Figure 2. It is hard to understand where this HMOE is working.
> > - Fourthly, from equation 6 and the related explanation, I am thinking the author misunderstood the concept of MoE. x and y are two different modalities, then the paper calls two encoders to process these two modalities are a mix of experts.
> > - Fifthly, I did not see any explanation about the hierarchical claim in Section 3.3, I did not understand why the "MoE" design is hierarchical.

---

> > > ### Author Response · Authors · 2025-11-27
> > > **Follow-up Response by Authors (Weakness 1 & 2)**
> > >
> > > **Q1. Clarification on the definition of Submodality**
> > >
> > > We clarify that the term submodality is not intended to redefine each token or property value as standalone modalities.
> > > Instead, we use it to denote semantically meaningful units within a modality, such as individual structural tokens in the PSMILES sequence and individual physicochemical attributes in the property vector.
> > >
> > > This level of granularity is consistent with established practice in chemical language modeling and polymer informatics:
> > > * ChemBERTa[1] explicitly applies token-level masking and reconstructs masked SMILES tokens using surrounding context (Section. 3.1), following the standard masked language modeling paradigm.
> > > * polyBERT[2] performs token-wise masked modeling on PSMILES strings (Section. polyBERT in Results), using individual tokens as the basic processing units for the structural sequence.
> > > * SPMM[3] treats the property vector as a sequence of elements and applies masking to individual property dimensions (Section. Handling SMILES and property values as a language in Methods), effectively operating at the level of per-property units.
> > > * PolyTAO[4] likewise processes properties as discrete units, converting each property into an individual token in the input prompt (Section. Prompt engineering in Methods)
> > >
> > > Collectively, these works show that structural tokens and individual property dimensions are commonly used as the basic units of processing in modern chemical language models and polymer informatics.
> > >
> > > Our use of the term submodality is therefore consistent with these conventions and simply denotes intra-modality units within our formulation, not a redefinition of the underlying modalities.
> > >
> > > [1] ChemBERTa: Large-Scale Self-Supervised Pretraining for Molecular Property Prediction (S. Chithrananda et al., 2020)
> > >
> > > [2] polyBERT: a chemical language model to enable fully machine-driven ultrafast polymer informatics (C. Kuenneth et al., 2023)
> > >
> > > [3] Bidirectional generation of structure and properties through a single molecular foundation model (Chang & Ye, 2024)
> > >
> > > [4] On-demand reverse design of polymers with PolyTAO (H. Qiu et al, 2024)
> > >
> > >
> > > **Q2. Explanation of the $\lambda^{xy}$ term in Eq. (6)**
> > >
> > > The third term $\lambda^{xy} \cdot p_{\theta}^{xy}(z \mid xy)$ in Eq. (6) introduced for theoretical completeness: it reflects the decomposition of the MI upper bound in Eq. (5), where $I_{\theta}(Z_{XY};XY)$ naturally appears in the analytical formulation. However, we intentionally set $\lambda^{xy}=0$ in all experiments.
> > >
> > > All downstream tasks in our evaluation require strictly unimodal inference, where the target modality is completely absent. Since the multimodal expert is never used at inference time, increasing its coefficient during training does not contribute to the objective of strengthening unimodal representations. Therefore, we assign the expert weights to $\lambda^{x}=\lambda^{y}=0.5$ (with $\lambda^{xy}=0$) to directly strengthen the informativeness of the modal-specific representations used at inference.
> > >
> > > Importantly, this choice does not eliminate the contribution of the cross-modal information. The InfoNCE objective aligns unimodal representations to the complete submodality representation, allowing them to indirectly leverage the cross-modal signal even when $\lambda^{xy}=0$.
> > >
> > >
> > > **Q3. Clarifying the HMoE in Figures**
> > >
> > > The HMoE is implemented inside the Transformer encoder and produces the representations for masked submodalities. In the revised manuscript, we explicitly indicate the HMoE within the Transformer encoder block in Figure 2 and clarify in the caption that the HMoE processes masked submodalities, whereas the Joint encoder processes the complete ones. This revision makes the role and placement of the HMoE clear.

---

> ### Author Response · Authors · 2025-11-21
> **Rebuttal by Authors**
>
> **W3. Misalignment between objective and experiment result**
>
> Firstly, we emphasize that the purpose of introducing the InfoNCE objective in MMPAE is not to improve robustness under missing-input conditions. Its primary role is to align the modality-specific representations with the complete submodality representation, thereby enforcing cross-modal alignment.
>
> Robustness to missing information instead mainly arises from the masked modeling components, which explicitly train the model to infer representations from all combinations of submodalities.
>
> For a fair comparison, all baselines were trained under the same masking protocol, allowing them to acquire similar robustness to incomplete inputs. Consequently, the performance gap between MMPAE and the baselines does not widen substantially as the masking ratio increases in Figures 3 and 4.
>
> Importantly, as shown in Appendix D, when the baselines are trained without this masking protocol and rely only on their original training strategies, the performance gap grows significantly as the missing input ratio increases. This demonstrates that the pattern observed in Figures 3 and 4 arises from the fact that all models were trained under the same masking protocol, rather than from any mismatch between our training objective and the empirical outcomes.
>
>
> **W4. Lack of ablation study**
>
> Our manuscript already provides ablations isolating the contributions of MMPAE, MMPAE+HMoE, and MMPAE+InfoNCE (Section 4). To further evaluate sensitivity to the objective design, the revised version adds ablations on the two major hyperparameters of MMPAE: the InfoNCE coefficient β and the temperature τ. Full ranges and results are included in Appendix F.
>
> The results show clear and consistent trends.
> * A moderate β offers a balanced trade-off between unimodal informativeness and cross-modal alignment. Very small β (e.g., 10) weakens alignment between modality-specific and complete representations, reducing the utility of unimodal latents, while excessively large β (e.g., 10000) over-dominates the objective and suppresses complementary modality-specific information.
>
> * The temperature τ exhibits a similar pattern. Performance is stable for τ in the range 0.1–0.3, whereas extreme values degrade alignment quality. Very small τ (0.05) produces overly sharp logits and weakens the contrastive signal, while large τ (0.5) reduces effective alignment strength.
>
> These ablations demonstrate that the proposed objective behaves predictably and that the improvements in Section 3 directly stem from the components introduced in our method.
>
>
> **W5. Issues in writing and presentation**
> The revised manuscript includes several improvements to the overall presentation. We have unified the fonts and formatting across all figures to ensure consistency with the main text. In addition, we have revised the writing to improve clarity and presentation for better readability.
>
>
> **Q1. Rationale for defining properties as modalities**
> Treating polymer properties as a separate modality is standard in polymer informatics because they arise from an information source fundamentally different from PSMILES. While PSMILES encodes monomer-level structural connectivity, the property vector captures macroscopic physicochemical characteristics determined by distinct underlying physical processes. Given this clear separation in information content and physical origin, modeling properties as a distinct modality is appropriate within a multimodal framework.
>
> This interpretation is aligned with recent multimodal polymer studies that explicitly handle structures and properties as separate modalities, using either numerical property vectors [1] or property-related textual descriptions [2–3]. Our formulation, which treats numerical properties as a modality complementary to PSMILES, follows these established practices.
>
> [1] Bidirectional generation of structure and properties through a single molecular foundation model (Chang & Ye, 2024)
>
> [2] Unified multimodal multidomain polymer representation for property prediction (Qi Huang et al., 2025)
>
> [3] PolyNC: a natural and chemical language model for the prediction of unified polymer properties (H. Qiu et al., 2023)

---

> > ### Comment · Reviewer_yNFG · 2025-11-26
> > **Reply to Response W3&4**
> >
> > - Thank the author for responding. Weakness 3 has been addressed.
> > - I still don't think the experiments of this paper are enough. At least the important ablation should be in the main paper instead of the Appendix. Section 4 spent most of the space describing the experiments; the real results are displayed in several figures and one small table. There should be more important results (eg, there are real-world experiments in the experiments), and ablations should be moved from the appendix to the main paper. That's why I raise this concern.
> > - The ablation studies not only mean changing some parameters and seeing the influence, but also **to prove the effectiveness of each key module in the main structure**, for instance, what will happen if you disable the HMoE? How will the performance change if you only use one modality? What if you disable the fusion but retain two modalities? Or what if we use more or less "sub-modalities"?
> > - Q1: The author either did not understand my question or did not directly answer the question. I have read the three papers the author referenced. [1] Use two modalities, one is the SMILES, and another is the property vector, and this paper treats the whole property vector of 53 properties as one modality; [2] Use 5 modalities, SMILES, graph, 3D, fingerprint, and text, none of which is related to this work. [3]  Also, the natural language and SMIELS modality, not related to this question. In conclusion, none of these papers defines each property as a sub-modality. These papers strengthen my opinion instead of the authors'.
> > - Q2: I know the decoding progress you are making. I am questioning the illustration of Figure 2, which is not consistent with your explanation.

---

> > > ### Author Response · Authors · 2025-11-29
> > > **Follow-up Response by Authors (Weakness 3 & 4)**
> > >
> > > We have updated the manuscript to incorporate all changes described in our follow-up response.
> > >
> > > Specifically, Section 4 has been reorganized to reduce descriptive overhead and to move key ablation studies and real-world evaluation results into the main paper, addressing the reviewer’s concern regarding the sufficiency and placement of experimental evidence.
> > >
> > > Additionally, the revised manuscript now includes the ablation that disables the HMoE while retaining the InfoNCE objective. This experiment has been added alongside the existing comparisons among the MMPAE variants, providing a more complete analysis of the contribution of each module.

---

> ### Author Response · Authors · 2025-11-21
> **Rebuttal by Authors**
>
> **Q2. Decoding using a single CLS representation**
>
> The CLS embedding is not used to directly generate the full PSMILES sequence. Instead, it serves as the conditioning latent vector for the autoregressive PSMILES decoder.
> As described in Sec. 3.1, we prepend a CLS token to the concatenated PSMILES–property token sequence, following the standard practice in Transformer-based architectures such as ViT [1] and BERT [2]. The CLS token summarizes information from the entire input sequence via self-attention, but it is not responsible for directly generating the entire sequence at once.
> During decoding, the model generates the sequence token-by-token under teacher forcing, using the ground-truth PSMILES embeddings as shown in Figure 1 (“GT PSMILES emb.”).
> Thus, the full PSMILES is reconstructed through a standard autoregressive decoding process conditioned on the CLS latent, rather than being produced directly from a single CLS token.
>
> [1] AN IMAGE IS WORTH 16X16 WORDS: TRANSFORMERS FOR IMAGE RECOGNITION AT SCALE (Alexey Dosovitskiy et al., 2021)
>
> [2] BERT: Pre-training of Deep Bidirectional Transformers for Language Understanding (Jaob Devlin et al., 2019)

---

> ### Comment · Reviewer_yNFG · 2025-11-26
> **Overall Comment**
>
> In the rebuttal stage, I don't think the author understands my concerns carefully and thoroughly. Most of my concerns are not addressed or not directly answered. Especially for the Q1 and W1, after carefully reading the reference papers, they directly prove my concerns instead of the author's claim about the sub-modality. And in Q2, the author refuses to admit there is a discrepancy between the figure and the explanation. Even more, there are still no key ablation experiments in the paper. These make me feel the authors have not carefully prepared the experiments and rebuttal. I remain more certain that this paper is not qualified. I will raise my confidence to reflect this.

---

> ### Author Response · Authors · 2025-11-27
> **Follow-up Response by Authors (Weakness 1 & 2)**
>
> **Q4. Clarifying the MoE definition in our formulation**
>
> We clarify that our use of the term Mixture of Experts (MoE) follows the formal definition widely adopted in the multimodal autoencoder literature. In this line of work, an expert is defined as the conditional probability distribution associated with a specific modality or subset, and an MoE is the weighted arithmetic aggregation of these experts, rather than by requiring multiple separate encoder networks.
>
> This definition is rigorously formalized in foundational works:
> * **MMVAE** [1] models the joint posterior as a mixture of unimodal posteriors, using the mixture-of-experts formulation to combine modality-specific conditional distributions:
>
>     $q_{\Phi}(z|x_{1:M}) = \sum_{m=1}^{M} \alpha_{m} \cdot q_{\phi_{m}}(z|x_{m})$, where each $q_{\phi_m}(z \mid x_m)$ is the modality-specific expert and $\alpha_m$ are mixture weights.
>
> * For **MVTCAE** [2], the authors formally categorize aggregation methods in Section 2.4. They explicitly define MoE as taking an “arithmetic mean of probability distributions” from view-specific encoders. They cite MMVAE as standard example of this MoE approach.
>
> Our formulation follows this same principle:
>
> - Eq. (1) : $p^J_\theta(z|xy) = \frac{1}{2^{|X|+|Y|}} \sum_{s \in \mathcal{P (xy)}}p^s_\theta(z|s)$  is a  uniform mixture of all subset experts.
> - Eq. (6) : $p_{\theta}^{HMoE}(z|xy) = \lambda^{x}p(z|x) + \lambda^{y}p(z|y) + \lambda^{xy}p(z|xy)$, is a hierarchical MoE over unimodal and multimodal experts.
>
> Thus, our usage of MoE is aligned with how the term is formally defined and used in multimodal autoencoder literature.
>
> [1] Variational Mixture-of-Experts Autoencoders for Multi-Modal Deep Generative Models (Y. Shi et al., 2018)
>
> [2] Multi-View Representation Learning via Total Correlation Objective (Hwang et al., 2021)
>
>
> **Q5. Clarifying the Hierarchical Structure of the MoE**
>
> Our MoE formulation is hierarchical because it consists of two distinct mixture layers:
> 1. **Low-level MoEs.**
>
> As shown in Eq. (4), we decompose the joint encoder into three experts: two unimodal experts (MoUE of X and MoUE of Y) and a multimodal expert (MoME of XY). Each expert corresponds to a mixture over the masked submodalities within its designated subset (X-only, Y-only, or cross-modal).
>
> 2. **High-level MoE.**
>
> These low-level experts are then combined through the gating weights  $\lambda^x$, $\lambda^y$, and $\lambda^{xy}$ to form the overall joint encoder (Eq (6)). This aggregation over the low-level MoEs constitutes the second layer of the hierarchy.
>
> Because the encoder is a mixture over (low-level) mixtures, the resulting structure is a two-level Mixture-of-Experts, which is why we refer to it as hierarchical. We have clarified this explicitly in the revised manuscript (Section 3.3.)

---

> ### Author Response · Authors · 2025-11-27
> **Follow-up Response by Authors (Weakness 3 & 4)**
>
> **Q1. Content organization and sufficiency of experiments section**
>
> To address this concern, we are currently reorganizing Section 4 to minimize descriptive setup and relocate key ablation studies and real-world results to the main paper. We will submit the revised manuscript containing these updates by **November 27, 23:59 PM (AoE).**
>
>
> **Q2. Necessity of comprehensive ablation studies of key modules**
>
> - Our manuscript already includes ablations on the key components of MMPAE. In particular, we report the stepwise comparison across MMPAE, MMPAE+HMoE, and MMPAE+InfoNCE, which explicitly isolates the contributions of the HMoE and the InfoNCE term. We will additionally include an ablation that disables the HMoE while retaining the InfoNCE objective, and update the results accordingly.  The additional results will be included in the revised manuscript by **November 27, 23:59 PM (AoE)**.
> - Regarding single-modality variants: the unimodal counterparts of our framework are provided through the Property Transformer (property-only) and Inverse Transformer (PSMILES-only) baselines, which correspond to structure-only and property-only prediction settings.
> - The ablation “disable the fusion but retain two modalities” is not compatible with our architecture. MMPAE utilizes an early-fusion mechanism, where a single Transformer jointly processes the concatenated PSMILES and property embeddings. Removing fusion would require a different architecture (e.g., late-fusion with separate encoders), which is outside the scope of our design. Furthermore, our downstream tasks require unimodal inference, so a variant that retains two modalities but removes fusion does not correspond to any inference scenario we aim to support and therefore is not a meaningful ablation for our problem setting.
> - Our framework can already flexibly handle “more or fewer visible submodalities” through the submodality-level masking mechanism.
> Because each submodality is masked independently, the model is naturally exposed to a wide range of partial-information settings. This makes additional manual manipulation of the submodality set unnecessary, as the masking distribution effectively covers such patterns within a unified formulation.
>
>
> **Q3. Clarification on the use of the term Submodality**
>
> Thank you for the clarification. We apologize for the confusion caused by our earlier response.
>
> First, we want to clarity the works [2–3] were cited not to claim that each property is a modality, but to support our rationale that properties collectively constitute a distinct modality separate from structural information.
> Uni-Poly[2] incorporates property-related textual descriptions as an additional modality that complements structural information. PolyNC[3] encodes property information (e.g., task-specific property descriptions) as natural language tokens, forming an input stream separate from chemical SMILES tokens. Taken together, these works indicate that property information can act as its own information source, distinct from structural inputs.
>
> Second, our use of the term submodality does not redefine each property value as its own modality.
> Instead, it denotes the heterogeneous components within a modality that we process independently.
> This usage follows the established practice in polymer and molecular language models:
>
> - SPMM[1] treats the property vector as a sequence of elements and applies masking to individual property dimensions (Section. Handling SMILES and property values as a language in Methods), effectively operating at the level of per-property units.
> - PolyTAO[4] likewise processes properties as discrete units, converting each property into an individual token in the input prompt (Section. Prompt engineering in Methods)
>
> Rather than redefining what constitutes a modality, our use of submodality provides a concise way to express the masking granularity required by our MI-based formulation and HMoE architecture.
>
> [1] Bidirectional generation of structure and properties through a single molecular foundation model (Chang & Ye, 2024)
>
> [2] Unified multimodal multidomain polymer representation for property prediction (Qi Huang et al., 2025)
>
> [3] PolyNC: a natural and chemical language model for the prediction of unified polymer properties (H. Qiu et al., 2023)
>
> [4] On-demand reverse design of polymers with PolyTAO (H. Qiu et al, 2024)
>
>
> **Q4. Inconsistency between Figure and Explanation**
>
> Figure 2 is intended as a conceptual illustration of how the reconstruction objective in Eq. (9) is combined with the InfoNCE alignment objective, rather than a detailed depiction of the autoregressive decoding pipeline. The actual decoding process is specified in Figure 1.
>
> In the revised manuscript, we have updated both Figure2 and its caption to to explicitly clarify this distinction and to eliminate the ambiguity raised by the reviewer.

---

### Official Review · Reviewer_QqP4 · 2025-10-31

**Soundness:** 2
**Presentation:** 2
**Contribution:** 2
**Rating:** 2
**Confidence:** 4

**Summary:**

In this work, the authors propose MMPAE, a multimodal autoencoder for polymer property prediction and inverse design. MMPAE is built on masked token prediction with regularization to balance information across different modalities. In particular, MMPAE includes an Transformer-based encoder, an autoregressive decoder for PSMILES generation, and an MLP for property prediction. The work also includes empirical study on property prediction and inverse design. It also shows advantage in cases with missing input compared with baseline models.

**Strengths:**

The proposed method attempts to build a unified model for both property prediction and inverse design, which is innovative.

**Weaknesses:**

1. Some details of empirical study is unclear.
2. The performance of proposed method doesn't show significant benefit over other baselines. For instance, in inverse design, MMPAE is on par with inverse Transformer as shown in Figure 4.
3. The model is trained and evaluated on PolyOne. However PolyOne is a synthetic dataset where properties are predicted by machine learning model, which makes it unclear about how the proposed method works on real-world tasks.

**Questions:**

1. In Figure 3, the missing input experiments may not faithfully reflect the scenario in reality. Though random masking PSMILES tokens may show the robustness of proposed method. In practice, it's usually not certain tokens that are missing but rather some chemical fragments that are totally missed.
2. $\beta$ in Eq.9 is set to 1000 which is very large. Does it mean that infoNCE loss dominates the training for MMPAE-InfoNCE?
3. Increasing training data size beyond 5M doesn't give much gain, does it mean the model is constraint by its capacity? Will a larger model be further improved with more data?
4. Can HMoE and InfoNCE be combined in training MMPAE?

---

> ### Author Response · Authors · 2025-11-21
> **Rebuttal by Authors**
>
> **W1. Unclear empirical details**
>
> We would like to clarify the key components of our empirical setup as follows.
>
> We use a batch size of 512, standardization for numerical properties (z-normalization), and a fixed 160-token representation for all PSMILES strings, as described in Appendix B. Models are optimized with AdamW (learning rate = 1e-4) under mixed-precision training (bf16). These settings are applied consistently across MMPAE variants.
> For baseline models, we evaluate two configurations: (i) their original hyperparameter settings as reported in the respective papers, and (ii) our unified training configuration described above. We report the better-performing result for each baseline to ensure a fair and performance-maximizing comparison.
>
> Further architectural details are provided in Section B.2 of the supplementary material. In addition, the complete training and evaluation code, including the training loop, data preprocessing pipeline, model implementation, and evaluation scripts, is publicly available in our repository:
> [https://github.com/MMPAE-ICLR2026/MMPAE.](https://anonymous.4open.science/r/MMPAE-4F3C/README.md)
>
>
> **W2. Limited performance improvement over baselines**
>
> The updated results in the revised manuscript reflect implementation refinements made after the initial submission. With this update, MMPAE demonstrates consistent improvements over all baselines across property prediction, inverse design, and cross-modal retrieval.
>
> These revised results more accurately represent the performance of the proposed framework and provide stronger empirical evidence for the benefit of jointly learning from both PSMILES and properties within a unified multimodal latent space, as described in Sec. 3.2.
>
>
> **W3. Real-world relevance of PolyOne**
>
> First, we would like to clarify the broader context of polymer informatics, in which experimentally measured polymer datasets are difficult and expensive to obtain at scale, and publicly available realistic datasets remain limited [1].
> Consequently, many studies in polymer informatics rely on simulation- or DFT-based datasets, and it is common to use model-generated or rule-based hypothetical datasets to enable large-scale training and benchmarking [2–4].
>
> To address the concern regarding real-world applicability, we additionally train and evaluate MMPAE and all baseline models on two realistic datasets, Point2 [5] and OpenPoly [6], both of which provide real polymer structures paired with experimentally measured properties.
> As is typical for experimentally curated polymer datasets, these datasets contain missing property values; we treat such missing entries as masked submodalities under our multimodal framework.
>
> The full evaluation under varying missing input conditions is provided in Appendix E.
> Despite the limited size and the presence of substantial missing values, MMPAE demonstrates strong and consistent performance across nearly all metrics. This indicates that the model effectively exploits multimodal interactions between PSMILES and properties, capturing complementary modality information.
>
> Overall, MMPAE exhibits strong performance on real polymer data and consistently outperforms baseline models across all tasks, addressing the concern about its practical applicability.
>
> [1] Machine Learning in Polymer Research (W. Ge., et al, 2025)
>
> [2] Generation of a large‑scale polymer library using rule‑based generative methods (Yue et al., 2024)
>
> [3] High‑Throughput Screening and Prediction of Polymers (T. Yue et al., 2023)
>
> [4] A graph representation of molecular ensembles for polymer property prediction (Aldeghi & Coley, 2022)
>
> [5] Point^2: A Polymer Informatics Training and Testing Database (J. Xu et al., 2025)
>
> [6] OpenPoly: A Polymer Database Empowering Benchmarking and Multi-property Predictions (J Wang et al., 2025)

---

> ### Author Response · Authors · 2025-11-21
> **Rebuttal by Authors**
>
> **Q1. Practical relevance of the PSMILES masking protocol**
>
> Our token-level masking evaluation is intended to measure the sensitivity of  a property prediction model to partially observable structural information. Masking individual PSMILES tokens offers a controlled mechanism for reducing structural cues, allowing us to assess whether the model can still infer relevant property signals under incomplete inputs. This setup aligns with the masked-token modeling protocols widely used in polymer sequence modeling [1-3].
>
> We note, however, that in practical scenarios, missing structural information often occurs at the chemical-fragment level rather than at individual tokens. To reflect a more practical and structured form of incompleteness, we additionally conduct a fragment-level masking evaluation using BRICS decomposition[4], where an entire chemically meaningful fragment is removed at random.
>
> | **Model**               | **RMSE (↓)** | **R² (↑)**   |
> |---------------------|----------|----------|
> | **MMPAE + InfoNCE**     | **0.1357**   | **0.9805**   |
> | Property Transformer | 0.1602  | 0.9661   |
> | SPMM                | 0.2974   | 0.9018   |
> | PolyBert            | 0.1987   | 0.9584   |
> | TransPolymer        | 0.2034   | 0.9537   |
>
> These results show that MMPAE maintains high predictive accuracy even under fragment-level masking, demonstrating that the model remains robust when chemically meaningful substructures are removed.
>
> We note that the other baselines also exhibit reasonable robustness under fragment-level masking. This is likely because the masking protocol exposes all models to diverse combinations of partially observed PSMILES submodalities, enabling them to retain informative structure cues even when chemically meaningful fragments are removed.
>
> Overall, these findings confirm that the token-level masking protocol remains reliable, as its conclusions are consistent with those observed under the more realistic fragment-level missing-structure setting.
>
> [1] polyBERT: a chemical language model to enable fully machine-driven ultrafast polymer informatics (Christopher Kuenneth & Rampi Ramprasad, 2023)
>
> [2] TransPolymer: a Transformer-based language model for polymer property predictions (C. Xu et al., 2023)
>
> [3] MMPolymer: A Multimodal Multitask Pretraining Framework for Polymer Property Prediction (F. Wang et al., 2024)
>
> [4] On the art of compiling and using 'drug-like' chemical fragment spaces (D Jorg et al., 2008)
>
>
> **Q2. Effect of a large β in Eq. 9**
>
> Yes. We intentionally set a large 𝛽 so that the InfoNCE loss becomes the dominant optimization signal in MMPAE-InfoNCE. The motivation is that InfoNCE explicitly aligns representations across submodalities, which is crucial for retrieval tasks where representations from different inputs are directly compared. In contrast, the reconstruction objective maximizes the same Mutual Information (MI) terms but cannot guarantee alignment, as the decoder can rely on many-to-one mappings that map distinct inputs to similar outputs.
>
> This design choice is supported by Figure 5, where adding InfoNCE (HMoE+InfoNCE) yields a clear improvement over the reconstruction-only variant (HMoE) in retrieval performance. We also observe consistent gains in PSMILES prediction (Figure 4) and property inference (Figure 3), suggesting that InfoNCE-driven alignment facilitates the training of decoders that remain robust under missing input submodalities.
>
> Further ablation studies on β are provided in Appendix F. As shown in these results, when β is small, the model assigns insufficient weight to aligning unimodal embeddings with the complete-submodality representation, leading to weaker performance in retrieval and related tasks. Increasing β strengthens this alignment and yields consistent improvements. However, when β becomes excessively large, the alignment objective begins to dominate the optimization and competes with reconstruction, particularly in property prediction, where overly strong contrastive alignment can reduce unimodal informativeness and slightly degrade performance under fully observed inputs. Overall, β = 10000 provides a stable balance, enhancing robustness without compromising reconstruction quality.
>
> These findings highlight that choosing an appropriate β is important: InfoNCE should guide alignment, but it must not overwhelm the complementary reconstruction objective.

---

> ### Author Response · Authors · 2025-11-21
> **Rebuttal by Authors**
>
> **Q3. Performance saturation and model capacity**
>
> The saturation observed beyond 5M samples does not indicate a limitation of model capacity. Instead, it reflects the limited marginal information in the PolyOne dataset. PolyOne is generated by combinatorial enumeration of a fixed fragment set, and its property values are produced by a machine-learning model. This results in a bounded chemical space and a restricted property distribution, so additional samples contribute minimal new structure–property signal.
>
> To assess whether saturation is due to the dataset rather than the model, we additionally evaluated MMPAE on two realistic experimental datasets, Point2 [1] and OpenPoly [2], which contain measured properties and substantially richer structural diversity. As reported in Appendix E, MMPAE achieves the strongest or second-strongest performance across inverse design, property prediction, and retrieval, consistently outperforming baselines. These results indicate that the lack of further improvement in PolyOne reflects the informational limits of the synthetic dataset rather than a bottleneck in model capacity.
>
> [1] Point^2: A Polymer Informatics Training and Testing Database (J. Xu et al., 2025)
>
> [2] OpenPoly: A Polymer Database Empowering Benchmarking and Multi-property Predictions (J Wang et al., 2025)
>
>
> **Q4. Joint training of HMoE and InfoNCE**
>
> Thank you for pointing out this aspect of the training setup. After the initial submission, we refined the implementation detail that prevented the HMoE reconstruction objective and the InfoNCE alignment objective from interacting as intended in our original formulation. We corrected this issue and re-evaluated all related experiments.
>
> With the updated implementation, jointly optimizing HMoE and InfoNCE leads to consistently stronger performance across property prediction, inverse design, and cross-modal retrieval. The revised manuscript has been updated accordingly, with the correct results and a clearer description of how HMoE and InfoNCE operate jointly in training.

---

### Official Review · Reviewer_VuYc · 2025-11-02

**Soundness:** 3
**Presentation:** 3
**Contribution:** 2
**Rating:** 6
**Confidence:** 3

**Summary:**

The authors present MMPAE, which is an autoencoder that allows to perform property prediction as well as structure generation
by unifying diverse polymer informatics. They also perform extensive experiments on large polymer datasets, showing the superiority of MMPAE.

The paper is well written and addresses important research problems on Polymer with novel technical details.
Additionally, the importance of MMPAE is empirically validated with several tasks including property prediction and polymer inverse design.

Two drawbacks are that no standard deviations are shown in the experimental results. Also, I am unsure of the importance of the metrics (i.e., validity, similarity and RMSE) used for the inverse design task, because these metrics do not always mean that MMPAE successfully design polymers of practical importance and because polymers generated by MMPAE are not shown at all.

Considering the pros and cons of the current paper, I recommend for a weak acceptance.

**Strengths:**

Technical novelty of MMPAE

Empirical results showing the promise

**Weaknesses:**

Analysis that does not include standard deviations

Unclarity of the significance of the metrics and the results on inverse design from a viewpoint of practice (e.g., actual usefulness of the polymer structures generated by MMPAE)

**Questions:**

What is a rational of the evaluation protocol of masking PSMILES for property prediction? Do you have any practical scenario of materials discovery where this protocol is useful?

---

> ### Author Response · Authors · 2025-11-21
> **Rebuttal by Authors**
>
> **W1. Lack of statistical reporting**
>
> We appreciate the reviewer’s concern. The revised manuscript now reports property-wise RMSE, relative RMSE, and R² for all 29 properties (Appendix H), which together quantify absolute accuracy, scale-normalized accuracy, and the proportion of variance explained by the model.
> We do not include standard deviation of prediction errors as a primary metric because it reflects only the dispersion of residuals and does not measure how well the model captures the underlying structure–property relationship. In regression settings, this relationship is more appropriately evaluated by R², which is the standard metric in polymer informatics.
> All reported metrics are computed in the original physical units of each property rather than the normalized values used during training to ensure physically meaningful interpretation.
>
> The property-wise results show that MMPAE+InfoNCE consistently achieves the lowest RMSE and relative RMSE and obtains high R² across nearly all properties, whereas baseline models exhibit larger errors and lower explained variance. Appendix I additionally provides property-wise scatter plots comparing predicted and ground-truth values for all 29 properties.
>
>
>
> **W2. Practical interpretability of inverse-design results**
> From a practical point of view, our inverse-design evaluation focuses on three metrics that directly correspond to the essential requirements of polymer design:
>
> 1. Validity[1]: Generated PSMILES must correspond to chemically plausible polymers: valence correctness, aromatic consistency, and polymerizability. This metric assesses whether the model proposes synthetically feasible structures rather than merely syntactically valid sequences.
>
> 2. Similarity[2]: Polymer properties are governed by monomer chemistry and repeat-unit structure, and it is well established in polymer science that polymers with similar structural motifs tend to exhibit similar property trends [3–4]. The similarity metrics reflects whether generated PSMILES lies in the correct chemical neighborhood analogous to the target properties.
>
> 3. RMSE [5]: In practical inverse design, a generated polymer is only useful if its properties match user-specified targets. RMSE directly quantifies the alignment between predicted and target properties and reflects how well the model satisfies the design objective.
>
> To further clarify the practical significance of these metrics, we provide qualitative examples in Appendix G.
>
> The generated PSMILES samples were produced by our trained MMPAE model using the property vector corresponding to each ground-truth polymer as input.
> In these examples, the polymers generated from the target property vectors exhibit high structural similarity and low RMSE relative to their corresponding ground-truth PSMILES. These results illustrate that the similarity metric reflects alignment at the level of meaningful chemical motifs, and that the generated structures recover property behaviors close to those of the intended targets.
>
> The generated candidates in each row share nearly identical backbone frameworks with closely matched side chain architectures. These structural elements are the primary determinants of key polymer properties including glass transition temperature, chain rigidity, segmental mobility, solubility and intermolecular interactions [6]. When the backbone structure such as the aromatic or heterocyclic ring system, chain connectivity and torsional constraints remains effectively unchanged the resulting thermal mechanical and electronic characteristics also fall within a comparable range [7]. The side chains in each pair exhibit similar branching patterns and functional group placement which preserve the solubility parameters packing tendencies and cohesive interactions [8]. As a result these candidates lie in a chemically coherent neighborhood where structural similarity reliably leads to similar bulk properties consistent with well established structure property relationships in polymer science. This demonstrates that the inverse designed structures are chemically meaningful and aligned with the intended property behavior rather than arbitrary outputs.
>
> [1] Automatic Chemical Design Using a Data-Driven Continuous Representation of Molecules (R Gomez-Bombarelli et al., 2018)
>
> [2] GuacaMol: Benchmarking Models for de Novo Molecular Design (N. Brown et al ., 2019)
>
> [3] Principles of Polymer Chemistry, Cornell University Press (Paul J. Flory, 1954)
>
> [4] Structure/chain-flexibility relationships of polymers (Z. Xu et al., 2005)
>
> [5] Machine learning enables polymer cloud-point engineering via inverse design (Kumar, J. N et al ., 2019)
>
> [6] Physical Properties of Polymers Handbook (James E. Mark, 2007)
>
> [7] Recent advances and challenges in experiment-oriented polymer informatics (Kan Hatakeyama-Sato et al., 2023)
>
> [8] The Role of the Side Chain on the Performance of N‑type Conjugated Polymers in Aqueous Electrolytes (A. Giovannitti et al., 2018)

---

> ### Author Response · Authors · 2025-11-21
> **Rebuttal by Author**
>
> **Q1. Rationale of the PSMILES masking evaluation**
>
> Our token-level masking evaluation is intended to measure how sensitive a property prediction model is to partial structure information. Masking individual PSMILES tokens provides a controlled way to reduce the amount of available structural information and to test whether a model can still infer relevant property signals under such incomplete input conditions. This aligns with the masked-token modeling objectives widely used in polymer sequence modeling [1-3].
>
> While the token-level masking setting provides a controlled way to assess robustness, structural incompleteness in practical scenarios more commonly arises at the *fragment* level. To reflect this, we additionally evaluate a more structured missing-information protocol using BRICS decomposition [4], which partitions a polymer into chemically meaningful fragments following established retrosynthetic disconnection rules. In this evaluation, we randomly remove one entire BRICS fragment from the PSMILES to simulate realistic fragment-level uncertainty.
>
> | **Model**               | **RMSE (↓)** | **R² score (↑)** |
> |---------------------|----------|--------------|
> | MMPAE + InfoNCE     | **0.1357**   | **0.9805**       |
> | Property Transformer | 0.1602   | 0.9661       |
> | SPMM                | 0.2974   | 0.9018       |
> | PolyBert            | 0.1987   | 0.9584       |
> | TransPolymer        | 0.2034   | 0.9537       |
>
> The results, included in Sec 4.4, show that MMPAE maintains stable performance under this more structured missing-structure condition, supporting the practical relevance of our evaluation, as polymer design workflows frequently involve uncertainty or variation at the fragment level.
>
> [1] polyBERT: a chemical language model to enable fully machine-driven ultrafast polymer informatics (Christopher Kuenneth & Rampi Ramprasad, 2023)
>
> [2] TransPolymer: a Transformer-based language model for polymer property predictions (C. Xu et al., 2023)
>
> [3] MMPolymer: A Multimodal Multitask Pretraining Framework for Polymer Property Prediction (F. Wang et al., 2024)
>
> [4] On the art of compiling and using ’Drug-Like’ chemical fragment spaces (J. Degen et al., 2008)

---

### Author Response · Authors · 2025-11-21
**General Response**

We sincerely appreciate all reviewers for their thoughtful and constructive feedback.

We have carefully examined each comment and provide detailed responses in the rebuttal. Below, we summarize the main concerns raised and how we addressed them.

The primary issues can be summarized as follows:

- **Practical relevance of the missing-input protocol (VuYc, QqP4):** We clarify the motivation behind our masking strategy and additionally provide fragment-level masking experiments that reflect realistic scenarios of incomplete structural information.
- **Real-world relevance of the PolyOne dataset (QqP4, QhSn)**: We explain the rationale for using PolyOne in the context of polymer informatics and report additional experiments on two realistic datasets (Point2 and OpenPoly), demonstrating that MMPAE maintains strong generalization under real experimental conditions.
- **Justification of the multimodal formulation (QhSn, yNFG):** We discussed why the polymer structure and properties are constitutes a valid multimodal setting in polymer informatics.
- **Novelty of the proposed method (yNFG)**: We emphasize that the main contribution lies not in architectural changes but in a principled information-theoretic objective that introduces a hierarchical mixture-of-experts formulation and an InfoNCE-based alignment to resolve a fundamental limitation of naive multimodal masking.
- **Practical interpretability of the inverse-design results (VuYc)**: We clarify the real-world meaning of the inverse-design metrics (validity, similarity, RMSE) and include qualitative examples accompanied by chemical interpretation to show that generated polymers are structurally coherent and property-aligned.

We are revised the manuscript to incorporate all reviewer feedback. This includes:

- Updating Section 3. to reflect implementation refinements that affected Eq. (9) and related explanations.
- Revising Section 4. to include updated experimental results and additional discussion for each task.
- Improving clarity and presentation for better readability.

If there are further concerns or suggestions, we welcome the opportunity to address them during the rebuttal period.

---

### Author Response · Authors · 2025-11-25
**Gentle Reminder**

Dear Reviewers,

We have updated the experimental results and revised the manuscript accordingly.

Together with the additional experiments and analyses provided in the rebuttal, we believe that the concerns raised by the reviewers have been addressed comprehensively.

We would be grateful if the reviewers could examine our responses and the revised results.

Any further feedback or remaining concerns would be highly valuable, and we remain fully committed to addressing them during the rebuttal period.

Sincerely,

Authors of Paper 24109

---

### Author Response · Authors · 2025-11-29
**General Response**

We have carefully revised the manuscript to strengthen our methodological justification and validation.

The key updates are summarized below:

- **Rationale for Property as Modality (Introduction)**: We motivated our treatment of numerical properties as a distinct modality by citing recent studies that adopt similar multimodal settings.
- **Restructured & Expanded Experiments (Section 4)**: We streamlined descriptive text to focus on key findings and incorporated a real-world benchmark and an ablation study to validate the critical components of our framework .
- **Detailed Analysis of Experimental Results (Appendix D):** We added a detailed analysis section to interpret the experimental results in depth, offering further insights into the underlying mechanisms.

We believe these revisions enhance the clarity and completeness of our work.

---

### Author Response · Authors · 2025-12-01
**Summary of Revisions and Responses**

We welcome the newly assigned Area Chair and sincerely appreciate your time and consideration.

Below, we provide a summary of our key contributions and the comprehensive revisions made to address the reviewers' concerns.

---

### **1. Key Contributions**

- We provide an information-theoretic analysis demonstrating that naive masked reconstruction creates an inherent **informativeness imbalance between modal-specific and cross-modal representations**  (Section 3.2) .
- We propose **MMPAE**, integrating a **Hierarchical Mixture-of-Experts (HMoE)** for explicit unimodal reweighting and an **InfoNCE** objective for robust cross-modal alignment (Section 3.3).
- We demonstrate that MMPAE achieves **consistently superior performance** across both the large-scale synthetic benchmark (**PolyOne**) and real-world datasets with inherent missing values (**Point²**; Section 4.2, **OpenPoly**; Appendix F) . Furthermore, we empirically verify the limitations of naive masking via rigorous ablation studies (Section 4.3) .


### **2. General Main Concerns and Our Responses/Revisions**

We believe that our rebuttal and revisions **substantively address all major concerns** raised by the reviewers.

**2.1 Validity of Multimodal Formulation (**QhSn, yNFG**)**

- **Concern:**
    - Reviewers questioned the validity of defining numerical properties and PSMILES as distinct modalities.
- **Response:**
    - We clarified that PSMILES (**structure**) and properties (**attributes**) originate from **heterogeneous sources**.
    - Treating them as distinct information sources is consistent with established multimodal frameworks.
- **Revisions:**
    - We updated the **Introduction** and **Related Work** to cite recent studies that explicitly validate treating numerical properties as independent modalities.


**2.2 Real world relevance of polyOne (**QqP4, QhSn**)**

- **Concern:**
    - Reviewers pointed out that the synthetic PolyOne benchmark may not sufficiently validate generalizability to real-world experimental scenarios.
- **Response and Revisions:**
    - To validate real-world applicability, we extended our evaluation to include **Point$\^2$** and **OpenPoly**, which consist of experimentally measured properties.
    - These results have been integrated into **Section 4.2** and **Appendix F.**
- **Key Findings:**
    - MMPAE maintains superior performance on these realistic benchmarks .
    - The widened performance margin in these data-scarce and missing-value settings confirms MMPAE effectively leverages complementary multimodal information


**2.3 Rationale of PSMILES Masking Protocol (**VuYc, QqP4**)**

- **Concern:**
    - Reviewers questioned the practical relevance of the random PSMILES masking protocol, suggesting it might not reflect realistic scenarios where entire chemical fragments are missing.
- **Response and Revisions:**
    - We clarified that token masking serves as a controlled mechanism to evaluate sensitivity to partial information.
    - To simulate realistic structural loss, we introduced a **BRICS fragment masking** experiment in **Appendix E.3**, where entire chemically meaningful substructures are removed .
- **Key Findings:**
    - MMPAE achieves superior performance under fragment masking without additional fine-tuning.
    - This confirms that our token-level protocol remains a valid and robust proxy for evaluating model performance under realistic fragment-level incompleteness.

### **3. Reviewer-Specific Concerns and Our Responses**

| Reviewer | Specific Concerns | Our Response and Revisions |
|----------|--------------------|-----------------------------|
| **VuYc** | Unclear practical significance of metrics and inverse design results. | We clarified the practical relevance of the metrics and added qualitative examples (App. H) showing generated structures preserve essential architectures aligned with targets. |
| **QqP4** | Questioned the rationale and sensitivity of the large $\beta$ coefficient. | We clarified $\beta$'s role in alignment and validated its effectiveness via ablation studies in App. G.1. |
| **yNFG** | Suggested improving the structure of Section 4 to prioritize experimental analysis and relocating key results to the main text. | We reorganized Sec. 4 to prioritize analysis over description and moved key ablations (e.g., InfoNCE w/o HMoE) to the main text to validate module effectiveness. |
| **yNFG** | Questioned the consistency between the InfoNCE objective and experimental trends regarding missing inputs. | We clarified that InfoNCE targets alignment and referenced App. E, showing baselines degrade significantly without our protocol. |
| **QhSn** | Suggested including recent multimodal methods as baselines (e.g., MMPolymer, Uni-Poly). | We clarified that suggested models require incompatible inputs (e.g., 3D/Graph) and explicitly justified their exclusion. |

---

### Meta-Review · Area_Chair_Bafk · 2026-01-06

**Summary:**

In this submission, the authors proposed a polymer representation method for jointly solving multiple polymer property prediction tasks. The proposed model leverages PSMILES strings and property vectors as input and considers a hierarchical MoE architecture. Experiments show the feasibility of the proposed method to some extent.

The main concerns of reviewers include three points: 1) it is not convincing that the proposed method is really multi-modal, 2) the usefulness of some key modules, e.g., the HMoE architecture, is not fully verified, and 2) the lack of some strong baselines. Although the authors made great efforts to revise the paper and add additional analytical experiments, some concerns remain unresolved.

In particular, after reading the paper and the discussions, I also challenge the necessity of the concept "sub-modality". The property vector works more like an auxiliary feature rather than an independent modality, and different properties, in my opinion, should not be treated as different (sub)modalities, either. In addition, without strong baselines like MMPolymer and Uni-Poly, the superiority of the proposed method is not convincing. Even if these methods use structural information and more modalities, they can still be treated as baselines for the solidity of the comparison experiments. Therefore, I think this work requires one more round of review.

**Reviewer Concerns:**

Although more experiments and explanations have been added in the revised paper, the concerns about the lack of strong baselines and the definition of the "multi-modal" are still outstanding.

**Reviewer Scores:**

I think the reviewers would have maintained their scores if they had participated fully in the discussion.

---

### Decision · Program_Chairs · 2026-01-26

Reject